# *SEMIC*: An efficient surface energy and mass balance model applied to the Greenland ice sheet

Mario Krapp[1,2], Alexander Robinson[3,1], and Andrey Ganopolski[1]

[1]Potsdam Institute for Climate Impact Research
[2]Department of Zoology, University of Cambridge
[3]Dpto. Astrofísica y CC de la Atmósfera, Universidad Complutense de Madrid

*Correspondence to:* mariokrapp@gmail.com

**Abstract.** We present *SEMIC*, a *S*urface *E*nergy and *M*ass balance model of *I*ntermediate *C*omplexity for snow and ice covered surfaces such as the Greenland ice sheet. SEMIC is fast enough for glacial cycle applications, making it a suitable replacement for simpler methods such as the positive degree day method often used in ice sheet modelling. Our model explicitly calculates the main processes involved in the surface energy and mass balance, while maintaining a simple interface and minimal data input to drive it. In this novel approach, we parameterise diurnal temperature variations in order to more realistically capture the daily thaw-freeze cycles that characterise the ice sheet mass balance. We show how to derive optimal model parameters for SEMIC specifically to reproduce surface characteristics and day-to-day variations similar to the regional climate model MAR (Modèle Atmosphérique Régional, version 2) and its incorporated multi-layer snowpack model SISVAT (Soil Ice Snow Vegetation Atmosphere Transfer). A validation test shows that SEMIC simulates future changes in surface temperature and surface mass balance in good agreement with the more sophisticated multi-layer snowpack model SISVAT included in MAR. With this paper, we present a physically-based surface model to the ice sheet-modelling community that is general enough to be used with in-situ observations, climate model or re-analysis data, and that is at the same time computationally fast enough for long-term integrations, such as glacial cycles or future climate change scenarios.

## 1  Introduction

Currently, surface melt accounts on average for about half of the observed Greenland ice sheet loss; the other half is lost through basal melt and ice discharge across the grounding line, i.e., calving (van den Broeke et al., 2009). Recent observations show that Greenland's surface mass balance is further declining (Hanna et al., 2013). The positive surface mass balance can no longer compensate losses via ice discharge and is therefore regarded as a dominant source of Greenland's total mass loss. The extreme melt season in 2012 exposed the Greenland ice sheet's vulnerability to long-lasting temperatures anomalies (Nghiem et al., 2012). As more marine terminating glaciers further retreat (Thomas et al., 2011), the partitioning of ice loss is likely to shift further towards the declining surface mass balance.

Numerical simulations of large land ice masses, such as the Greenland and the Antarctic ice sheets, require numerical models to be fast because the response time of ice sheets to changes in the surface mass balance is slow, on the order of years to tens of

millennia (Cuffey and Paterson, 2010). Hence, many thousands of years of model integration are required to spin-up the model or to simulate one or several glacial cycles.

The simplest, fastest, and still most widely used method to estimate the surface mass balance of glaciers and ice sheets is the so-called positive-degree-day (PDD) approach (e.g. Reeh, 1991; Ohmura, 2001). It is based on the empirical relationship between surface melt rate and daily mean surface air temperature. Although PDD parameters are tuned to correctly represent present-day melting rates, past climates may require different parameter values. For instance, the PDD approach with its present-day parameter values is not applicable to orbitally-forced climate change (van de Berg et al., 2011; Robinson and Goelzer, 2014).

The importance of climatic changes in the past to the sensitivity of the Greenland ice sheet has already been acknowledged in one of the first attempts to utilise an energy-balance model for a sensitivity study (Oerlemans, 1991). Under a warming climate an energy-balance approach is superior to the relatively simple PDD method and "snowpack properties evolve on a multidecadal timescale to changing climate, with a potentially large impact on the mass balance of the ice sheet" (Bougamont et al., 2007).

Here, we propose a physically-based model utilising an energy balance approach that is inherently consistent with a variety of climate states different from today, e.g., future warming, last glacial maximum, or the Eemian interglacial. Our proposed model not only accounts for temperature changes but also for changes in other climate factors, such as insolation, turbulent heat fluxes, and surface albedo.

The *Surface Energy and Mass balance model of Intermediate Complexity* (SEMIC) is based on a surface scheme that has already been used to study glacial cycles (Calov et al., 2005). SEMIC provides a process-based relationship between surface energy and surface mass balance changes. The approach described here guarantees a consistent treatment of melting and melt water refreezing; both are important processes for the mass budget of ice sheets (Reijmer et al., 2012).

Compared to more sophisticated multi-layer snowpack models, which include snow metamorphism or vertical temperature profile calculations (e.g., Vionnet et al., 2012), SEMIC has a reduced complexity, one-layer snowpack. This saves computation time and allows for integrations on multi-millennial time scales. SEMIC calculates the daily surface energy and mass balance throughout the year but is also fast enough to focus on longer time scales when climatological changes determine the trend of the surface energy and mass balance.

Numerical ice sheet models need the annual mean surface temperatures and annual mean surface mass balance of ice as boundary conditions at the surface. Both are calculated by SEMIC, which can thus be directly coupled to the ice sheet model. There is a multitude of possible applications for SEMIC, for example, under projections of future warming for the next centuries or glacial cycle simulations. In this paper, we will discuss the future warming projections of the RCP8.5 scenario (Moss et al., 2010) to demonstrate the capabilities of our model.

The paper is organised as follows. In the next section, we present the model equations and their parameters. In Sect. 3, we describe the calibration procedure used to constrain the free model parameters and we estimate the sensitivity of the calculated surface mass balance with respect to the model parameters. In Sect. 4, we validate our model against regional climate model data for a future warming scenario. We discuss our findings in Sect. 5 and conclude in Sect. 6.

With this paper we acknowledge, support, and encourage research that follows standards with respect to scientific reproducibility, transparency, and data availability. The model source code and the authors' manuscript source is freely available and accessible online.

## 2  Model Description

SEMIC is based on the calculation of the mass and energy balance of the snow and/or ice surface (see, for example Greuell et al., 2004). We assume that the surface temperature $T_s$ responds to changes in the surface energy balance according to

$$c_{\text{eff}}\frac{dT_s}{dt} = (1-\alpha)SW^{\downarrow} + LW^{\downarrow} - LW^{\uparrow} - H_S - H_L - Q_{\text{M/R}} \tag{1}$$

where $\alpha$ is the surface albedo, $SW^{\downarrow}$ is the downwelling shortwave radiation; $(1-\alpha)SW^{\downarrow}$ is the net shortwave radiation $SW_{\text{net}}$. $LW^{\downarrow}$ is the downwelling longwave radiation, $LW^{\uparrow}$ is the upwelling longwave radiation, $H_S$ and $H_L$ are the sensible and latent heat flux to the atmosphere, and $Q_{\text{M/R}}$ the energy flux related to phase transitions, i.e., for melting or refreezing of snow and ice. The parameter $c_{\text{eff}}$ denotes the effective heat capacity of the snowpack. In a strict sense of the term "energy balance" the left-hand-side of Eq. (1) should be zero. Here, we assume that surface temperature and the energy are not in equilibrium because the snowpack or surface exerts some thermal inertia.

Temperatures of snow- and ice-covered surfaces cannot exceed 0° C. However, for computational purposes, we initially assume that $T_s$ represents the potential temperature, which would be observed in the absence of phase transitions, i.e., melting or refreezing. Once, melting and refreezing has been computed (see Sect. 2.3), the residual heat flux $Q_{\text{M/R}}$ in Eq. (1) keeps track of any heat flux surplus or deficit and is added back to the energy balance. This way, $T_s$ never exceeds 0° C over snow and ice.

For coupling to an ice sheet model, the surface mass balance for ice ($SMB_i$) is computed by SEMIC. It separates the total surface mass balance into the surface mass balance for snow and for ice:

$$SMB = SMB_s + SMB_i = P_s - SU - M + R \tag{2}$$

$$SMB_s = P_s - SU - M_{\text{snow}} - C_{si} \tag{3}$$

$$SMB_i = C_{si} - M_{\text{ice}} + R \tag{4}$$

Here, $P_s$ is the snowfall rate and $SU$ is the sublimation rate which is related to the latent heat flux via $H_L/\rho_w L_s$, with $\rho_w$ and $L_s$ being water density and latent heat of sublimation, respectively (see Table 1). The model variable $M$ is the total melting rate, i.e., the sum of snow and ice melt (denoted by the subscripts), $R$ is the refreezing rate of liquid water (rain or melt water), and $C_{si}$ is the compaction rate of snow which is turned into ice.

Changes in snowpack height $h_s$ (in meter water equivalent) are determined by the surface mass balance of snow:

$$\frac{dh_s}{dt} = SMB_s, \quad \text{with} \quad h_s \in \max(0, h_{s,\text{max}}) \tag{5}$$

If the snow height $h_s$ exceeds a certain threshold $h_{s,\mathrm{max}}$ (here set to $5\,\mathrm{m}$) snow is transformed into ice—in a simple way resembling snow compaction:

$$\int_0^{\Delta t} C_{si}\,dt = \max(0, h_s - h_{s,\mathrm{max}}) \tag{6}$$

The described equations are solved using an explicit time-step scheme with a time step of one day. In principle, the use of monthly input data is also supported but would require interpolation to daily time steps.

## 2.1 Surface heat fluxes

We describe the outgoing longwave radiation as a function of surface temperature according to the Stefan–Boltzmann law:

$$LW^{\uparrow} = \sigma T_s^4 \tag{7}$$

For the turbulent heat exchange (sensible and latent) we use a standard bulk formulation (e.g., Gill, 1982):

$$H_S = C_S \rho_a c_{p,a} u_s (T_s - T_a) \tag{8a}$$

$$H_L = C_L \rho_a L_s u_s (q_s - q_a) \tag{8b}$$

with sensible and latent heat exchange coefficients $C_S$ and $C_L$, air density $\rho_a$, specific heat capacity of air $c_{p,a}$, surface wind speed $u_s$, air temperature $T_a$, latent heat of sublimation/deposition $L_s$, and air specific humidity $q_a$. Air density $\rho_a$ is not available from MAR and thus approximated by the ideal gas law $\rho_a = \frac{p}{R_s T_a}$, with specific gas constant $R_s = 258\,\mathrm{J\,kg^{-1}\,K^{-1}}$, and surface pressure $p$, which is available from MAR. Specific humidity over the snow or ice surface ($q_s$) is assumed to be saturated and depends on surface pressure $p_s$ and saturation water vapour pressure $e^*$

$$q_s = \frac{e^* \epsilon}{e^*(\epsilon - 1) + p_s}, \quad \text{where} \tag{9}$$

$$e^* = 611.2 \exp\left(a \frac{T_s - T_0}{T_b + T_s - T_0}\right)$$

with $\epsilon = 0.62197$, the ratio of the molar weights of water vapour and dry air, and coefficients $a$, $T_b$, which are prescribed for vapour pressure over water ($a = 17.62$, $T_b = 243.12\,\mathrm{K}$) or ice/snow ($a = 22.46$, $T_b = 272.62\,\mathrm{K}$). $T_0$ denotes the freezing point of water, $273.15\,\mathrm{K}$. See, Gill (1982) for more details.

## 2.2 The diurnal cycle of thawing and freezing

Because we use daily time steps, processes on time scales shorter than one day cannot be resolved explicitly. Hence, we cannot explicitly account for the thawing during daytime and the freezing during nighttime which is quite usual for the melting season on Greenland. The absorbed shortwave radiation, for example, can exhibit large diurnal variations, especially when the surface albedo is low (Cuffey and Paterson, 2010). During the day, near surface temperatures may rise above freezing temperature and

snow or ice starts to melt. During the night, temperatures drop below freezing and any liquid water such as previously melted water can refreeze within the snowpack.

To account for this process we introduce a parameterisation for the diurnal cycle of thawing and freezing. We simply assume a sinusoidal temperature curve $T(t)$ throughout the day (here, units of time $t$ are hours h) around a given mean surface
temperature $T_s$ (here, we refer to $T_s$ with units in °C) with amplitude $A$, i.e., a cosine function (Fig. 1a):

$$T(t) = T_s - A\cos(\frac{2\pi}{24}t) \tag{10}$$

For the sake of simplicity we use a single constant $A$, although in reality it is spatially and temporally dependent as shown in Fig. 1b.

Melting and refreezing may then occur on the same day if (potential, not actual) $T_s$ exceeds 0°C. The amount of melting and
refreezing then depends on the amplitude $A$ and the mean daily temperature $T_s$ (Fig. 1a). Fortunately, an analytical solution to this problem exists. We calculate the roots of the cosine function and then integrate between the roots to solve for average above- and below-freezing mean surface temperatures $T_s^+$ and $T_s^-$. The roots are

$$t_1 = \frac{24}{2\pi}\arccos(\frac{T_s}{A}) \quad t_2 = 24 - t_1.$$

Thus, the time span for temperatures above and below freezing is

$$\Delta t_+ = t_2 - t_1 = 24 - 2t_1, \quad \text{and} \quad \Delta t_- = 2t_1.$$

This leads us to an expression for averages of above- and below-freezing temperatures $T_s^+$ and $T_s^-$. These are the integrals of the cosine function

$$T_s^+ = \frac{1}{\Delta t_+}\int_{t_1}^{t_2} T(t)dt \tag{11a}$$

$$= \frac{24}{\pi\Delta t_+}\left[-T_s\arccos(\frac{T_s}{A}) + A\sqrt{1 - \frac{T_s^2}{A^2}} + \pi T_s\right]$$

$$T_s^- = \frac{1}{\Delta t_-}\int_0^{t_1} T(t)dt + \int_{t_2}^{24} T(t)dt \tag{11b}$$

$$= \frac{24}{\pi\Delta t_-}\left[T_s\arccos(\frac{T_s}{A}) - A\sqrt{1 - \frac{T_s^2}{A^2}}\right].$$

This parameterisation depends on on the prescribed diurnal cycle amplitude, $A$, which affects the amount of melting and refreezing and, thus, the surface mass balance. Note, melt energy $Q_m$ and "cold content" $Q_c$ in the following Eq. (12) are
calculated by using $T_s^+$ and $T_s^-$, respectively. Without this parameterisation or with $A$ set to zero, melting and refreezing cannot occur at the same time step and instead, the actual surface temperature $T_s$ must be used.

## 2.3 Melting and refreezing

Additional processes that affect the snowpack temperature are melting and refreezing. During the course of one day the energy available for melt $Q_m$ and refreezing (the so-called "cold content") $Q_c$ are defined as

$$Q_m = \begin{cases} (T_s^+ - T_0)\frac{c_{\text{eff}}}{\Delta t} & \text{if } T_s^+ > T_0, \\ 0 & \text{if } T_s^+ \leq T_0, \end{cases} \tag{12a}$$

and

$$Q_c = \begin{cases} 0 & \text{if } T_s^- \geq T_0, \\ (T_0 - T_s^-)\frac{c_{\text{eff}}}{\Delta t} & \text{if } T_s^- < T_0. \end{cases} \tag{12b}$$

Thus, the potential melt is

$$M_{\text{pot}} = \frac{Q_m}{\rho_w L_m} \tag{13}$$

with latent heat of melting (or fusion) $L_m$, and time step $\Delta t$. Actual melt depends on how much snow or ice is available for melt. If potential melt is larger than the current snow height all snow melts down and the excess melt energy is used to melt the underlying ice. Ice-free land is treated differently and the excess melt energy is used to warm the surface. The actual melt $M$ is then the sum of melted snow and melted ice:

$$M_{\text{snow}} = \min(M_{\text{pot}}, h_s/\Delta t) \tag{14a}$$

$$M_{\text{ice}} = M_{\text{pot}} - M_{\text{snow}} \tag{14b}$$

$$M = M_{\text{snow}} + M_{\text{ice}} \tag{14c}$$

The refreezing rate depends on the potential liquid water to be refrozen, i.e., the actual melt rate $M$ and rainfall $P_r$. Analogous to the melt rates, the potential refreezing is given by

$$R_{\text{pot}} = \frac{Q_c}{\rho_w L_m}. \tag{15}$$

Suppose some rain or melt water exists within the snow pack. The "cold content" $Q_c$ is then used to (virtually) turn this liquid water into frozen water, i.e., snow or ice. We distinguish between refrozen rain and refrozen melt water

$$R_{\text{pot,rain}} = \min(R_{\text{pot}}, P_r) \tag{16a}$$

$$R_{\text{pot,melt}} = \min(\max(R_{\text{pot}} - R_{\text{pot,rain}}, 0), M_{\text{snow}}) \tag{16b}$$

$$R = R_{\text{rain}} + R_{\text{melt}} = f_R(R_{\text{pot,rain}} + R_{\text{pot,melt}}) \tag{16c}$$

Because of its porous structure the snowpack retains a limited amount of melt water and this melt water retention is reflected by the refreezing correction parameter $f_R$ which acts on the potential refreezing of rain and melt water. In contrast, ice itself does

not retain any melt water at the surface, so we assume that it has a water holding capacity of zero. We can therefore neglect refreezing of melted ice and treat ice melt as runoff.

As noted in the beginning of this section, melting consumes internal energy of the snowpack, while refreezing releases internal energy. SEMIC accounts for both melting and refreezing, and therefore the associated temperature change in Eq. (1) via $Q_{M/R}$—the residual energy for refreezing or melting:

$$Q_{\text{M/R}} = \rho_w L_m (M - R) \tag{17}$$

Here, we see how tightly the mass balance and the energy balance are coupled and that great care must be taken when the underlying surface processes are incorporated into one model.

## 2.4 Snow albedo parameterisation

We use a simple surface albedo parameterisation that depends on the snow albedo, the background albedo, and the snow height (Oerlemans and Knap, 1998). The surface albedo $\alpha$ defined as is the average of fresh snow albedo $\alpha_s$ and the prescribed background albedo $\alpha_i$ for ice-covered or $\alpha_l$ for ice-free land and depends on the critical snow height $h_{\text{crit}}$

$$\alpha = \alpha_s - \exp\left(\frac{-h_s}{h_{\text{crit}}}\right)(\alpha_s - \alpha_{bg}) \quad \text{where} \quad \alpha_{bg} = \begin{cases} \alpha_i & \text{for ice-covered or} \\ \alpha_l & \text{for ice-free land} \end{cases} \tag{18}$$

We also compared our approach to a more sophisticated albedo parameterisation that includes a temperature-dependent snow albedo (Slater et al., 1998) but concluded that the added value is too little given the reduction in model performance.

## 2.5 Model Setup

To drive the model we need as input: incoming short- and longwave radiation, near-surface air temperature, surface wind speed, near-surface specific humidity, surface pressure, snowfall, and rainfall, either computed by a coupled atmosphere model or prescribed as atmospheric forcing. Forcing fields are listed in Table 2.

We use daily mean data from the regional climate model MAR, version 2 (Fettweis et al., 2013), which includes the multi-layer snowpack model SISVAT (Soil Ice Snow Vegetation Atmosphere Transfer), to tune and optimise our model parameters. At its lateral boundaries MAR is forced by the general circulation model CanESM2 under historical conditions and under the global warming scenario RCP 8.5 (for details, see Fettweis et al., 2013). As input to SEMIC, we use the MAR output from the historical period, i.e., 10 years from 1990–1999, and from the 21st century scenario RCP8.5, i.e., 10 years from 2090–2099, as these periods represent present-day climate and future extreme warming conditions for the Greenland ice sheet well.

To reduce the large amount of forcing data for the whole 20 years, we simply use a random sub-sample accounting for 25% of land and ice points. The overall memory demand for the calibration procedure is thus reduced by a factor of about 10. For each new initialisation, the model requires several years of spin-up—especially the snow pack height $h_s$ and hence the associated surface albedo $\alpha$ (see Eq. 18) responds rather slowly. Therefore, we loop 10 times over each of the 10-year periods

to advance the variables from their initial conditions. The output from the last iteration, i.e., the final 10 years, is then used for the comparison with MAR output.

Our current setup is designed to allow testing and tuning of the snowpack model driven by prescribed atmospheric forcing. Thus feedbacks with the atmosphere via near-surface heat fluxes are currently not active, reducing the degrees of freedom of the model. It is important to remember that while SEMIC is driven by atmospheric forcing from MAR, the main comparison is with MAR's snowpack model SISVAT although SEMIC calculates several surface-atmosphere heat fluxes such latent heat, sensible heat, and upward longwave radiation as done by MAR. But for the sake of clarity, from now on we refer to MAR whenever a comparison between SEMIC and MAR/SISVAT output is being made.

On a modern laptop (e.g., MacBook Pro with an Intel Core i7, 2.8 GHz), 100 years of integration with daily time steps on a grid with 6,720 points (i.e., the MAR grid with 25 km horizontal resolution) take about 40 seconds for SEMIC. Of course, in coupled and stand-alone applications there is overhead for exchanging the variables and writing the output, thus, adding to the overall computation time. However, SEMIC is a fast model and therefore well suited for multi-millennial integration such as glacial cycles.

## 3 Model parameter calibration

To calibrate our free model parameters we minimise errors with respect to MAR output. Afterwards the optimised parameters are used to compare SEMIC with results for the whole historical period from 1970–2005 and for the warming scenario RCP 8.5 from 2006–2100. The periods 1990–1999 and 2090–2099 represent a subset, i.e., a training data set of the historical period and the RCP8.5 scenario.

At the model initialisation, $T_s$ and $\alpha_s$ are prescribed with values from MAR output of the first days, i.e., Jan 1 1990 and 2090. Because we do not know the water equivalent snow height from MAR and we initially set $h_s = 1\,\mathrm{m}$. After a few time steps the fast responding variables $T_s$ and $\alpha_s$ are close to their expected trajectories. However, response time for $h_s$ is much longer and difficult to quantify because it depends on the slowly varying and highly sensitive mass balance terms. Therefore, several years of integration can be necessary for the model spin-up. To account for the longer response time of $h_s$ we loop 10 times over the 10 years, 1990–1999 and 2090–2099, creating an effective integration period of 100 years. From those 10 loops, the final loop over the 10 years is used to estimate the error between SEMIC and MAR. The model initialisation and spin-up is done every time SEMIC uses a new model parameter set, in order to treat each of those parameter settings in a comparable way.

The quality of our parameters is measured with the normalised centred root mean square error $E$. It is a good way to estimate how closely a test field (SEMIC output in our case) resembles a reference field (MAR output) in terms of correlation and variance (Taylor, 2001) while also allowing the assessment of variables with different units:

$$E = \sqrt{\frac{1}{N}\sum_{n=1}^{N}\left[\frac{(X_n - \overline{X}) - (Y_n - \overline{Y})}{\sigma_Y}\right]^2 + \left[\frac{\overline{X} - \overline{Y}}{\sigma_Y}\right]^2} \tag{19}$$

Here, $X$ is some SEMIC time series with $N$ time steps. This could be any model variable, for example, averaged surface temperature $T_s$, net shortwave radiation $SW_{net} = (1-\alpha)SW^\downarrow$, or surface mass balance $SMB = P_s - SU - M + R$. The symbol $Y$ represents the corresponding MAR time series and the $\sigma$'s are the standard deviations of the time series. Overbars denote temporal averages of the time series.

## 3.1 Minimising the cost function

To include Greenland's diverse climate zones, we choose the time series (i.e., the $X_n$'s and $Y_n$'s) as being spatial averages over ice-free land and over three different ice-covered regions, all shown in Fig. 2. The three ice-covered regions crudely represent the main ablation zones at the ice-sheet margins (region 1), the main accumulation zone at ice-sheet interior (region 3), and a mixed zone in between the main accumulation and ablation zones (region 2). Note, the outlined regions represent different mass balance zones for today's climate and may change for any future warming scenario such as RCP8.5. Still, the distinction is useful to derive a differentiated response in each of those regions to the atmospheric forcing. We calculate four different $E$ values, one over ice-free land ($E_L$) and three over the different ice-covered regions ($E_{b1}, E_{b2}, E_{b3}$) for both periods, 1990–1999 and 2090–2099, denoted by a subscript, e.g., $E_L^{hist}$ or $E_{b2}^{rcp85}$.

For our cost function we regard the following variables as important for the surface energy and mass balance: surface temperature $T_s$, net shortwave radiation $SW_{net}$, melt $M$, and surface mass balance $SMB$. The magnitude of this vector then defines our cost function $J$

$$J = \left\| \left( E_{L,T_s}^{hist}, E_{b1,T_s}^{hist}, \ldots, E_{L,SW_{net}}^{hist}, \ldots, E_{b3,SMB}^{hist}, E_{L,T_s}^{rcp85}, \ldots, E_{b3,SMB}^{rcp85} \right)^T \right\| \tag{20}$$

which we want to minimise. Note that we assign different area weights to each of the regions.

The cost function $J$ is minimised with a method called *Particle Swarm Optimisation*, described below. Using these calibration steps, we derive these optimal parameters values: $A = 3.0\,\mathrm{K}$, $\alpha_s = 0.79$, $\alpha_i = 0.41$, $\alpha_l = 0.07$, $h_{crit} = 0.028\,\mathrm{m}$, and $f_R = 0.85$ which are also listed in Table 3.

## 3.2 Particle Swarm Optimisation

Because of the high dimensionality of the parameter space, a random search for the optimal parameters would need a large sample size in the order of $\mathcal{O}(10^{5-6})$. One optimisation technique that overcomes the problem of large sample sizes is the so-called *Particle Swarm Optimisation* (PSO) (Poli et al., 2007). PSO is based on social interaction among particles of the 'swarm'. Initially, each particle is placed randomly in the parameter space and has a random velocity. For all particles the cost function $J$ is calculated (Eq. (20)) . This determines the "fitness" of each individual and of the swarm as a whole. Now, each particle updates its current position and velocity in the parameter space depending on its current and current-best fitness position, and also on the global best-fitness position, with some random perturbations. The next iteration starts after all particles have moved. Eventually, the swarm as a whole moves to the minimum of the cost function $J$. For our parameter calibration we

let 30 particles freely swarm within the four-dimensional parameter space. The global best-fitness solution found within 100 iterations[1] is then regarded as optimal.

## 3.3 Calibration results

The ice-sheet surface temperature is very well constrained by the atmospheric forcing fields. Therefore, the surface temperature in SEMIC is similar to the one calculated by MAR, as the annual mean differences and the ice-sheet averaged time series show (Figs. 3, 4, 5, and 6). The annual mean difference between SEMIC and MAR for years 1990–1999 (2090–2099) is about $0.4\,\mathrm{K}$ ($0.3\,\mathrm{K}$) over the ice sheet and $0.5\,\mathrm{K}$ ($0.2\,\mathrm{K}$) over ice-free land. While large parts of the ice sheet are colder in SEMIC, temperatures at the ice divides and over ice-free land are generally warmer in SEMIC (see Figs. 3i and 4i).

The surface mass balance is well captured by SEMIC. The largest differences occur in the ablation zones of region 1 and 2 around the margin of the ice sheet. While melting[2] over the northern part of the ice sheet is overestimated by SEMIC, it is underestimated over the southern part of the ice sheet. Nonetheless, for years 1990–1999 (2090–2099) the overall surface mass balance difference over the ice sheet between SEMIC and MAR is almost zero, -0.04 (-0.03) $\mathrm{mm\,day^{-1}}$, with SEMIC having an average surface mass balance of 1.57 (-0.24) $\mathrm{mm\,day^{-1}}$, and 1.61 (-0.21) $\mathrm{mm\,day^{-1}}$ in MAR. SEMIC and MAR also exhibit similar melt rates over the ice sheet with differences of -0.06 (-0.15) $\mathrm{mm\,day^{-1}}$. A detailed overview of the differences from the model variables that we used to define the cost function is provided in Table 4.

In regions where surface mass balance is positive (see Fig. 3c and g and 4c and g), errors are small because accumulation is mainly prescribed by snowfall and to a lesser extent by sublimation/evaporation. Therefore, differences in ablation are more important because they arise dynamically from SEMIC. The introduced diurnal cycle parameterisation is critical here; it allows melting and refreezing within one time step which would be prohibited otherwise.

SEMIC is able to capture both the increase and decrease of surface mass balance as well as the seasonal melting as shown for the different regions and periods in Figs. 5 and 6. As can be seen from Fig. 7, errors in melt rates and the surface mass balance accumulate over time. The calibration procedure minimises discrepancies across the four regions and across the two different calibration periods. This results in melt rates that are slightly too large in all regions and for both periods but the surface mass balance itself is reasonably well modelled by SEMIC except for the inner ice sheet region 3 for the years 2090–2099. Overall, using the resulting optimal parameters from the calibration improves SEMIC's performance in modelling the whole historical and RCP8.5 period from 1970–2099 as shown in the next Sect. 4.

The Taylor diagram in Fig. 9 summarises the performance of SEMIC compared to MAR's multi-layer snowpack model. Except the surface mass balance for the RCP8.5 years and the melt for the historical period in the interior of the Greenland ice sheet (region 3), all variables are reasonably close to the reference value of each regions' time series in terms of their variability, measured via their standard deviation and their match to the corresponding MAR variables, measured via their correlation. A detailed look into each time series (Fig. 5 and 6) further supports our results that SEMIC and MAR variables are reasonably close to each other, especially during the whole melt season.

---

[1]Note, 100 iterations are a pre-defined upper limit and usually solutions tend to converge earlier.
[2]Note that melt is defined here as a positive quantity but is subtracted from the surface mass balance.

The overall differences between SEMIC and MAR temperature and surface mass balance are small given the challenge of i) matching both periods, 1990–1999 and 2090–2099, ii) calibrating different mass and energy balance variables in parallel, and iii) using only a subset of grid points (25%) averaged over four regions across entire Greenland. SEMIC's annual mean values of surface temperature and surface mass balance are well suited for applications of interactive ice sheet models. The optimisation guarantees that the regionally averaged MAR and SEMIC time series are as close as possible (as defined by the cost function). Still, SEMIC is sensitive to the choice of parameters, so we now show how perturbed parameters around their optimal values affect the surface energy and mass balance of the ice sheet.

## 3.4 Parameter Sensitivity

We identified parameters that dominate model uncertainties and tested the parameter sensitivity on the model performance (e.g., Fitzgerald et al., 2012). We addressed the sensitivity of the SEMIC model parameters listed in Table 3 by varying each parameter freely while keeping the others fixed at their optimal value. In this way, we estimated the contribution of each individual parameter on the cost function $J$.

As can be seen for all parameter sensitivity graphs in Fig. 8, the Particle Swarm Optimisation was able to find an optimal parameter set for which the PSO minimises $J$. Therefore, we are confident that this optimal parameters set provides us with a globally optimised model setup.

The cost function shows a large sensitivity to variations of the diurnal cycle amplitude $A$ and the fresh-snow albedo $\alpha_s$. The sensitivity to the other albedo-relevant parameters, that is $\alpha_i$, $\alpha_l$, and $h_{\mathrm{crit}}$ is rather small. The diurnal cycle and thus $A$ directly affect melt and the surface mass balance. The local minimum of the cost function for $A$ is also in line with the range of diurnal cycle amplitude values around the ablation zone of the Greenland ice sheet, as is modelled by MAR during summer (Fig. 1b).

The parameter $\alpha_s$ directly affects the radiation budget, where a small percent change makes a large difference in terms of receiving short-wave radiation. The cost function is less sensitive to the other albedo parameters. Values of $h_{\mathrm{crit}}$ below $2\,\mathrm{cm}$ or above $5\,\mathrm{cm}$ would lead to non-optimal solutions because it dictates how much ice and how much snow can be "seen" by short-wave radiation and, in this way, influences the surface energy balance.

The optimal refreezing correction parameter $f_R$ is 0.85 (see Table 3). This large proportion of melt water refreezing underlines the importance of the refreezing process in determining the surface mass balance of the Greenland ice sheet. Any lower refreezing correction leads to a less optimal cost function.

Having determined the optimal parameter set we can now compare SEMIC with MAR for the whole historical and RCP8.5 period from 1970–2100.

## 4 Model validation

As a final step of the full model analysis, we use the optimised model parameters for the following two model validation runs: a) A historical run from 1970–2005 and b) an RCP8.5 scenario run from 2006–2100. This time, we compare SEMIC with MAR for a whole time series instead of just a few years as done for the calibration. We take a closer look into the regional differences

of surface temperature, surface melt, and surface mass balance over the four previously defined regions and calculate the corresponding time series of their annual mean values, as shown in Fig. 10.

Annual mean surface temperatures correspond well with MAR results and both time series are hard to distinguish from each other. To a lesser extent but still reasonably well, surface melt and surface mass balance are captured by SEMIC. The decline of surface mass balance throughout the 21st century in the RCP8.5 scenario is evident over the three ice-sheet regions, while the mass balance remains close to zero over ice-free land. Furthermore, SEMIC captures the year–to–year variations throughout the historical and the RCP8.5 period. This tells us that the newly introduced diurnal cycle parameterisation makes SEMIC more realistic and thus comparable to more comprehensive and complex multi-layer snowpack models. We believe that a representation of the diurnal thawing and freezing cycle is essential for SEMIC and for physically correct mass balance modelling in general, and thus represent an important advance.

The overall performance of SEMIC with respect to the more sophisticated regional climate model MAR is satisfactory, given its intended use for long time-scale simulations. In the validation test we show that SEMIC is able to capture long-term trends of the Greenland ice sheet under the RCP8.5 scenario, while also reproducing the interannual variability exhibited by MAR.

## 5  Discussion

In this study we describe the new intermediate complexity snowpack model SEMIC, and compare its performance to a state of the art model. As the main use for SEMIC would be for long time-scale simulations of the of ice sheets, we focus on simulating the surface mass balance of the Greenland ice sheet for the present and future under a strongly changing climate. For this purpose, comparing with regional climate model results is most informative. It should be noted, however, that SEMIC can be used to simulate any type of snowpack, as long as the forcing variables are available for driving the model. This includes other regional climate models such as RACMO (Noël et al., 2015), re-analysis data such as ERA-Interim (Dee et al., 2011), and even in-situ observational data sets such as PROMICE (van As et al., 2016). In fact, a preliminary analysis (which is beyond the scope of this study) using meteorological data from Col de Porte (Morin et al., 2012) suggests that SEMIC is also capable to reproduce reasonable results when forced by observational data. Yet, for a more comprehensive validation, we used output from MARv2 forced by CanESM2 under the RCP8.5 scenario, as described in Franco et al. (2013) and which Xavier Fettweis has made publicly available. While MARv2 has been superseded by MARv3.5.2 (Fettweis et al., 2016), we expect that the results of our tuning exercise would not change significantly using either version. More importantly, one benefit of SEMIC is that it is computationally fast and lends itself to ensemble experiments that do not rely on one guess of the parameter values.

The definition of a cost function for the model calibration is a non-trivial task. SEMIC computes several variables which, in principle, could all be included in the cost function. We choose to take into account, first, the net shortwave radiation which is determined by the albedo parameterisation and its parameters and which in turn determines surface temperatures. Second and third, the surface mass balance and the surface temperature are considered, in anticipation of the interactive coupling to an ice-sheet model. And fourth, melting to account for the newly introduced diurnal cycle parameterisation of thawing and freezing. Still, it is clear that the choice of the cost function and the variables considered is subjective.

In the model calibration and validation we weighted each of the regions on the area. The area of the ice-free land and region 1, for example, is nearly as large as either region 2 or 3. Consequently, the influence of the smaller regions—here, land and region 1—is much smaller than that of the larger ones, such as regions 2 or 3, despite region 1 being a major driver of surface melting.

For the calibration of model parameters, we chose ten years at the end 20th century, i.e., years 1990–1999 from the historical period and ten years at the end of the 21st century, i.e, years 2090–2099 from the RCP8.5 scenario. Those years cover periods of moderate melt under present-day climate conditions and more extreme melt under a strong warming scenario. Forcing SEMIC with both moderate and extreme climate conditions shows that our model is capable of representing the surface energy and mass balance of the Greenland ice sheet under different climate conditions and is thus very well suited for future and past
climate studies such as glacial cycles.

There are two main reasons why surface temperature is better represented in SEMIC than the surface mass balance: 1) Surface temperature is determined by the driving atmospheric processes, which in our case are prescribed by MAR atmospheric forcing. Therefore changes in the atmosphere are directly reflected at the surface in terms of energy balance. 2) Surface mass balance is harder to constrain because the processes within the snowpack are more complex. Mass can be added by the
atmosphere via rain and snowfall, and mass can be removed via melting. Within the snowpack melted water can refreeze if the temperature allows that. Refreezing depends on the available liquid water, i.e., rain or melted ice/snow, and on the energy budget, i.e., the "cold content". The multitude of feedbacks involved in the surface mass balance makes it far less constrained by external forcing variables than surface temperature.

We only describe the large-scale effects of changes in the snowpack and we omit a microscopic description of snow physics
(e.g., Vionnet et al., 2012). SEMIC can therefore be thought of as a surrogate of a more complex multi-layer snowpack model. We have developed SEMIC as a coupler between interactive ice sheet models and EMICs (Earth-System Models of Intermediate Complexity) or coarse resolution GCMs (General Circulation Models). SEMIC realistically represents the energy transfer between atmosphere and surface as radiation and turbulent mixing of heat and water vapour, thus providing a general solution to the surface energy balance that is applicable for different climates and time scales.

Ice-free land and ice-covered land are treated differently in SEMIC because of the different physical processes involved. For example, the surface temperature of ice- and snow-free land has no upper limit as is the case for surface temperatures of ice, which is always lower than or equal to the freezing point. Generally, land albedo is much more variable than as described by the single bare land albedo used in SEMIC. Different land and vegetation types have different effects on the radiation budget. Consequently, net shortwave radiation errors in SEMIC are larger over ice-free land than over the ice sheet (Figs. 3j and 4j).
Details in model representation also reveal differences between SEMIC and MAR. However, these differences are not so much related to the underlying physical principles, i.e., the assumption of energy and mass balance of the snow- and ice-covered surface, as to the choice of parameters made in order to match SEMIC variables to MAR variables.

SEMIC makes use of two simple but effective parameterisations that are important for its good performance: One is the surface albedo for which we already discussed the problem of the net shortwave radiation budget over ice-free land. Although
the net shortwave radiation has an effect on the surface energy balance, errors do not translate directly into errors in the surface

temperature (Figs. 3i and 4i). One reason is that the contribution of sensible and latent heat flux is larger over ice-free land because of the larger temperature contrast. Latent heat flux, for example, is about 10 times larger over ice-free land than over the ice sheet.

Another reason for SEMIC's good performance is the newly introduced diurnal cycle parameterisation, which allows for faster computation while adding the daily thaw–freeze cycle during melt season. The representation of the diurnal cycle of the whole ice sheet by a single constant value is somewhat problematic because in reality, it changes over time and location, depending on the climatic conditions, e.g., cloud cover and its effect on downwelling longwave radiation. Still, the overall results of SEMIC with respect to surface mass balance are satisfactory. The diurnal cycle opens many new aspects which could improve model results, e.g., a spatial dependence such as height-dependent amplitude or a direct calculation of the amplitude by the coupled atmospheric model, but this is beyond the scope of this paper. Also, a different or a more realistic albedo scheme could replace the current simple albedo parameterisation (Oerlemans and Knap, 1998). SEMIC has also been successfully tested with a temperature-dependent albedo scheme (Slater et al., 1998), for example.

Our results underpin the consistent representation of the dominant processes involved in the complex interactions between snow- or ice-covered surfaces and the atmosphere. SEMIC incorporates simpler dynamics compared to multi-layer snowpack models, but represents the essential surface energy and mass balance processes, and is still fast in terms of computational time.

SEMIC is well suited for long-term integrations up to several millennia and has been successfully tested for the last 78,000 years (data taken from Heinemann et al., 2014, personal communication). From the 100 year run-time estimate we can assume that computation of the surface mass balance on every single day during one glacial cycle (of about 100 k years) would take about 11 h. Current state-of-the-art multi-layer snowpack models are not able to perform such long integrations but they also do not serve this purpose. Under these circumstances, using a much simpler model—such as SEMIC—is advised.

SEMIC is well suited for applications with global climate models which have just started to master glacial time scales (e.g., Heinemann et al., 2014). SEMIC will be part of the next version of the regional energy and moisture and balance model REMBO (Robinson et al., 2010) and is also ready to be coupled to an interactive ice-sheet model. SEMIC is considered as an open-source project, therefore contributions are welcome, and we encourage and support the integration of SEMIC into climate and ice-sheet models.

## 6   Conclusions

We have presented a new *S*urface *E*nergy and *M*ass balance model of *I*ntermediate *C*omplexity (SEMIC) for snow- and ice-covered surfaces that is simple and fast enough for long-term integrations up to glacial time scales. SEMIC is a physically based model that accounts for energy and mass balance and it can be used as a surrogate for computationally intensive regional climate models with their multi-layer snowpack models. The most important features of SEMIC are a simple but effective surface albedo parameterisation and a parameterisation of the daily thaw-freeze cycle that allows partitioning between melting and refreezing. SEMIC has been forced with atmospheric fields from the regional climate model MAR (MARv2) and compared to MAR's multi-layer snowpack model SISVAT, SEMIC represents surface temperature and surface mass balance considerably

well. For the RCP8.5 warming scenario, SEMIC correctly simulates the climatological trend and the interannual variability of surface temperature and the mass balance of the ice sheet. SEMIC hereby incorporates a minimum number of free model parameters and a large effort was made to balance the complexity of the represented processes in favour of faster computation.

## 7 Scientific Reproducibility, Transparency, and Data Availability

We hereby acknowledge, support, and encourage research that follows standards with respect to scientific reproducibility, transparency, and data availability. Any model source code and the authors' manuscript source (typeset in LaTeX) is freely available and accessible online.

The project infrastructure covering individuals step starting from data download and preparation, model source code compilation, running the optimisation, running the calibrated model, running the model with historical and RCP8.5 scenario data, as well as the source code of this manuscript with its figures can be downloaded from the repository website https://gitlab.pik-potsdam.de/krapp/semic-project. See the project website's `README.md` for details. The project can also be cloned using `git`:

```
git clone -b v1.1 git@gitlab.pik-potsdam.de:krapp/semic-project.git
```

The atmospheric forcing data from the MAR/CanESM2 model for the historical period from 1970–2005 and for the RCP8.5 scenario for the period from 2005–2100 are available at ftp://ftp.climato.be/fettweis/MARv2/.

*Author contributions.* M.K., A.R., and A.G. designed the model. M.K. implemented the model code with contributions from A.R.. M.K. implemented and carried out the model calibration and the data analysis. M.K. prepared the mansucript with contributions from all co-authors.

*Competing interests.* The authors declare that they have no conflict of interest.

*Acknowledgements.* We would like to thank Xavier Fettweis for providing MAR/CanESM2 data. M.K. is also grateful to Malte Heinemann and Axel Timmermann for their kind hospitality during his research visit at the *International Pacific Research Center* (SOEST, University of Hawaii). A.R. was funded by the Marie Curie 7th Framework Programme (Project PIEF-GA-2012-331835, EURICE). M.K. was funded by the Deutsche Forschungsgemeinschaft (DFG) Project "Modeling the Greenland ice sheet response to climate change on different timescales".

| symbol | value | description |
|---|---|---|
| $\Delta t$ | $86,400\,\mathrm{s}$ | time step of one day |
| $c_{\mathrm{eff}}$ | $2 \cdot 10^6\,\mathrm{J\,m^{-3}}$ | effective heat capacity snow/ice (volumetric) |
| $C_S$ | $2.0 \cdot 10^{-3}$ | sensible heat exchange coefficient |
| $C_L$ | $0.5 \cdot 10^{-3}$ | latent heat exchange coefficient |
| $c_{p,a}$ | $1,000\,\mathrm{J\,kg^{-1}K^{-1}}$ | specific heat capacity of air |
| $\sigma$ | $5.67 \cdot 10^{-8}\,\mathrm{W\,m^{-2}K^{-4}}$ | Stefan–Boltzmann constant |
| $T_0$ | $273.15\,\mathrm{K}$ | freezing point of water |
| $\rho_w$ | $1,000\,\mathrm{kg\,m^{-3}}$ | density of liquid water |
| $L_s$ | $2.83 \cdot 10^6\,\mathrm{J\,kg^{-1}}$ | latent heat of sublimation |
| $L_v$ | $2.5 \cdot 10^6\,\mathrm{J\,kg^{-1}}$ | latent heat of vaporisation |
| $L_m$ | $3.3 \cdot 10^5\,\mathrm{J\,kg^{-1}}$ | latent heat of melting $(L_s - L_v)$ |
| $h_{s,\mathrm{max}}$ | $5.0\,\mathrm{m}$ | maximum snow height (cut-off) |

**Table 1.** Model constants and their description.

| symbol | description |
|---|---|
| $SW^{\downarrow}$ | downwelling shortwave radiation [$\mathrm{W\,m^{-2}}$] |
| $LW^{\downarrow}$ | downwelling longwave radiation [$\mathrm{W\,m^{-2}}$] |
| $\rho_a$ | air density [$\mathrm{kg\,m^{-3}}$] |
| $u_s$ | surface wind speed [$\mathrm{m\,s^{-1}}$] |
| $T_a$ | near-surface air temperature [K] |
| $q_a$ | near-surface specific humidity [$\mathrm{kg\,kg^{-1}}$] |
| $p_s$ | surface pressure [Pa] |
| $P_s$ | snowfall rate [$\mathrm{m\,s^{-1}}$] |
| $P_r$ | rainfall rate [$\mathrm{m\,s^{-1}}$] |

**Table 2.** Atmospheric forcing fields needed as input for this model.

| symbol | range | value | description |
|---|---|---|---|
| $A$ | 0.0–5.0 | **3.0** | amplitude of diurnal cycle [K] |
| $\alpha_s$ | 0.70–0.90 | **0.79** | fresh dry snow albedo |
| $\alpha_i$ | 0.25–0.55 | **0.41** | bare ice albedo, i.e., clean or blue ice |
| $\alpha_l$ | 0.05–0.35 | **0.07** | bare land albedo |
| $h_{\mathrm{crit}}$ | 0.00–0.20 | **0.028** | critical snow height for albedo parameterisation [m] |
| $f_R$ | 0.0–1.0 | **0.85** | refreezing correction |

**Table 3.** Model parameters with their initial range and their optimal value in bold face.

**Table 4.** Comparison of SEMIC and MAR. Shown are multi-year mean averages over the ice sheet (regions 1–3) and ice-free land, their mean gridpoint-to-gridpoint differences $\overline{\Delta}$, their minimum, and their maximum gridpoint-to-gridpoint differences, $\min\Delta$ and $\max\Delta$. Here, ice sheet means all ice-covered regions (region 1–3).

| | | 1990–1999 | | | | | 2090–2099 | | | | |
|---|---|---|---|---|---|---|---|---|---|---|---|
| | | SEMIC | MAR | $\overline{\Delta}$ | $\min\Delta$ | $\max\Delta$ | SEMIC | MAR | $\overline{\Delta}$ | $\min\Delta$ | $\max\Delta$ |
| ice sheet | $T_s$ [K] | 249.6 | 249.2 | 1.4 | 0.2 | 4.8 | 256.1 | 255.8 | 1.3 | 0.4 | 3.7 |
| | $SMB$ [mm day$^{-1}$] | 1.57 | 1.61 | 0.96 | -1.78 | 2.88 | -0.24 | -0.21 | 0.97 | -2.40 | 4.70 |
| | $M$ [mm day$^{-1}$] | 1.62 | 1.68 | 0.94 | -0.79 | 3.68 | 4.05 | 4.20 | 0.84 | -2.64 | 4.20 |
| | $SW_{\text{net}}$ [W m$^{-2}$] | 28.7 | 27.7 | 1.9 | -10.9 | 14.2 | 31.9 | 32.0 | 0.9 | -8.7 | 10.8 |
| land | $T_s$ [K] | 258.4 | 257.9 | 1.5 | -0.1 | 5.1 | 267.5 | 267.3 | 1.2 | 0.0 | 3.2 |
| | $SMB$ [mm day$^{-1}$] | 1.27 | 1.25 | 1.03 | 0.67 | 1.56 | 1.09 | 1.00 | 1.09 | 0.98 | 1.87 |
| | $M$ [mm day$^{-1}$] | 2.18 | 2.04 | 1.14 | 0.60 | 1.79 | 2.37 | 2.25 | 1.12 | 0.64 | 1.44 |
| | $SW_{\text{net}}$ [W m$^{-2}$] | 46.8 | 47.3 | 0.4 | -20.7 | 22.6 | 61.7 | 65.6 | -2.9 | -13.5 | 8.3 |

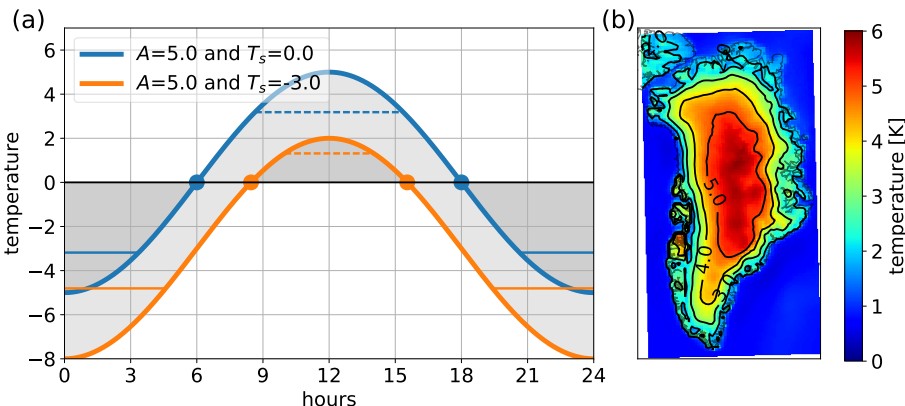

**Figure 1.** The diurnal cycle parameterised as cosine function with amplitude $A$ around the mean temperature $T_s$ (a). The dashed horizontal line marks the analytical solution of the average above-mean temperature $T_s^+$ and the solid horizontal lines mark the below-mean temperature $T_s^-$ (see Eq. 11a and b). The circles denote the roots of the sinusoidal temperature cycle curve. The mean diurnal cycle amplitude of air temperature for the summer season (JJA) in MAR for the years 1990–1999 (b).

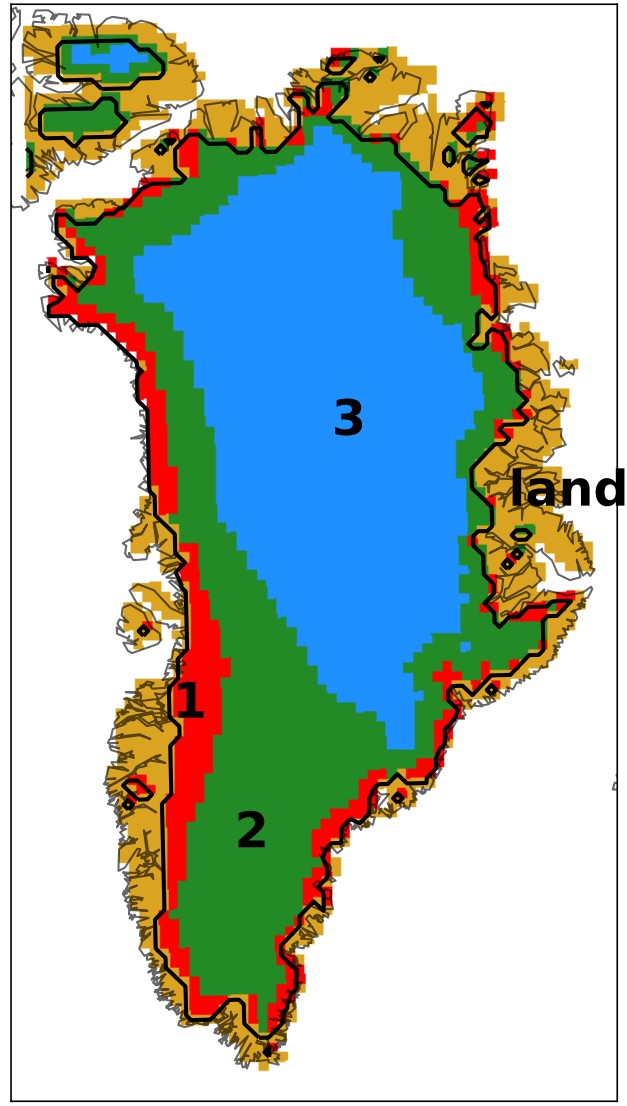

**Figure 2.** This region mask is used to estimate the region-averaged time series for the model calibration. Region 1 represents the ice margin, while the other regions represent areas with seasonal melt (2) or almost no melt (3). This mask is readily available from the MAR model data (named MSK). Note that these regions are only representative for present-day climatic conditions, in a strict sense. However, in a broader sense, we regard them also as useful to differentiate the future warming climatic response such under the RCP8.5 scenario.

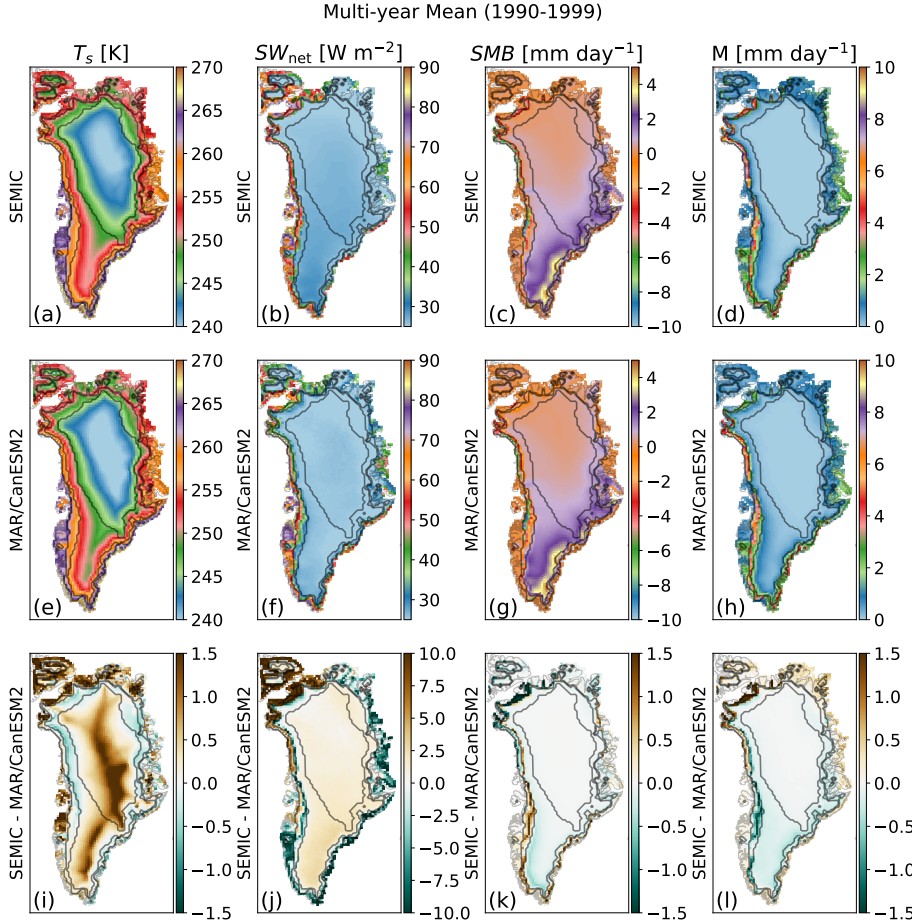

**Figure 3.** Comparison of multi-year (1990–1999) mean surface temperature $T_s$, net shortwave radiation $SW_{net}$, surface mass balance $SMB$, and surface melt $M$ as modelled by SEMIC (a)–(d) and MAR (e)–(h) and the differences between SEMIC and MAR (i)–(l). The outlined contours show the boundaries of the three ice-covered MAR regions as shown in Fig. 2. See Table 4 for values of minimum and maximum differences.

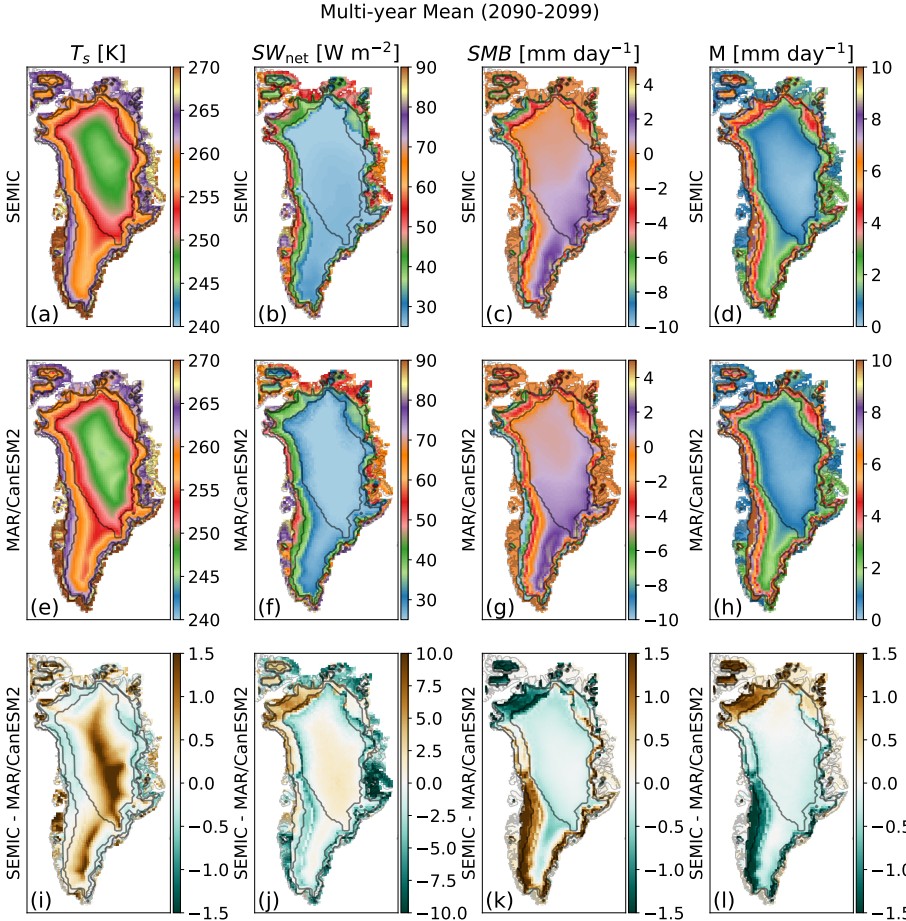

**Figure 4.** Comparison of multi-year (2090–2099) mean surface temperature $T_s$, net shortwave radiation $SW_{net}$, surface mass balance $SMB$, and surface melt $M$ as modelled by SEMIC (a)–(d) and MAR (e)–(h) and the differences between SEMIC and MAR (i)–(l). The outlined contours show the boundaries of the three ice-covered MAR regions as shown in Fig. 2. See Table 4 for values of minimum and maximum differences.

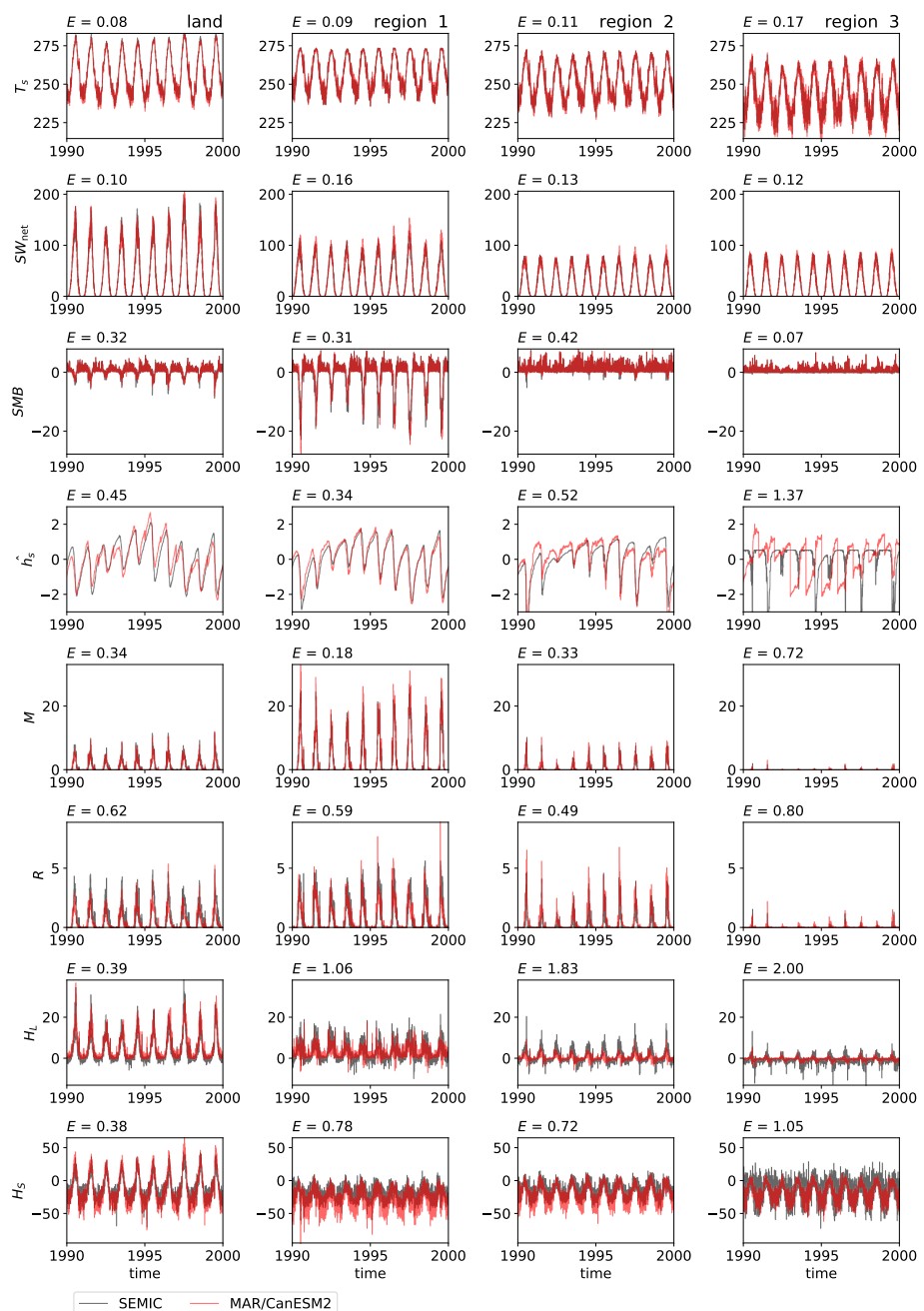

**Figure 5.** Time series of ice–sheet averaged surface temperature $T_s$ [K], net shortwave radiation $SW_{\mathrm{net}}$ [W m$^{-2}$], surface mass balance $SMB$ [mm day$^{-1}$], standardised snow height $\hat{h_s}$, surface melt $M$ [mm day$^{-1}$], refreezing $R$ [mm day$^{-1}$], latent heat flux $H_L$ [W m$^{-2}$], and sensible heat flux $H_S$ [W m$^{-2}$] as calculated by MAR and by SEMIC with optimal parameters from Table 3 for the years 1990–1999 of the historical period. Note that $h_s$ is scaled via its standard deviation because SEMIC and MAR incorporate a different criterion of maximum snow height (5 m in SEMIC; more than 10 m in MAR). The annotated number on the top left of each frame is the computed centred root mean square error as defined in Eq. (19) and it marks the distance to the reference field as shown in the Taylor diagram Fig. 9a.

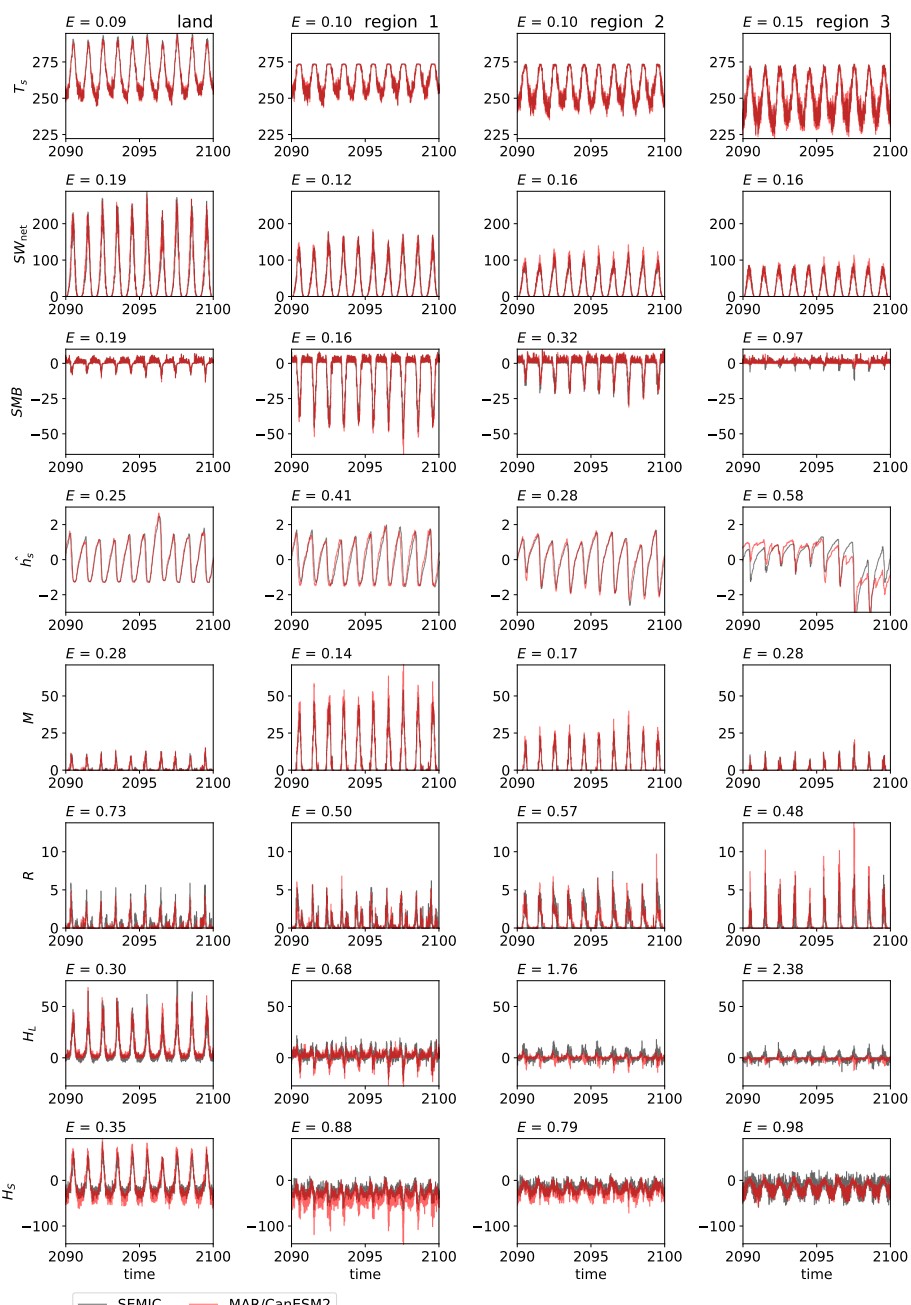

**Figure 6.** Time series of ice–sheet averaged surface temperature $T_s$ [K], net shortwave radiation $SW_{\mathrm{net}}$ [W m$^{-2}$], surface mass balance $SMB$ [mm day$^{-1}$], standardised snow height $\hat{h_s}$, surface melt $M$ [mm day$^{-1}$], refreezing $R$ [mm day$^{-1}$], latent heat flux $H_L$ [W m$^{-2}$], and sensible heat flux $H_S$ [W m$^{-2}$] as calculated by MAR and by SEMIC with optimal parameters from Table 3 for the years 2090–2099 of the historical period. Note that $h_s$ is scaled via its standard deviation because SEMIC and MAR incorporate a different criterion of maximum snow height (5 m in SEMIC; more than 10 m in MAR). The annotated number on the top left of each frame is the computed centred root mean square error as defined in Eq. (19) and it marks the distance to the reference field as shown in the Taylor diagram Fig. 9b.

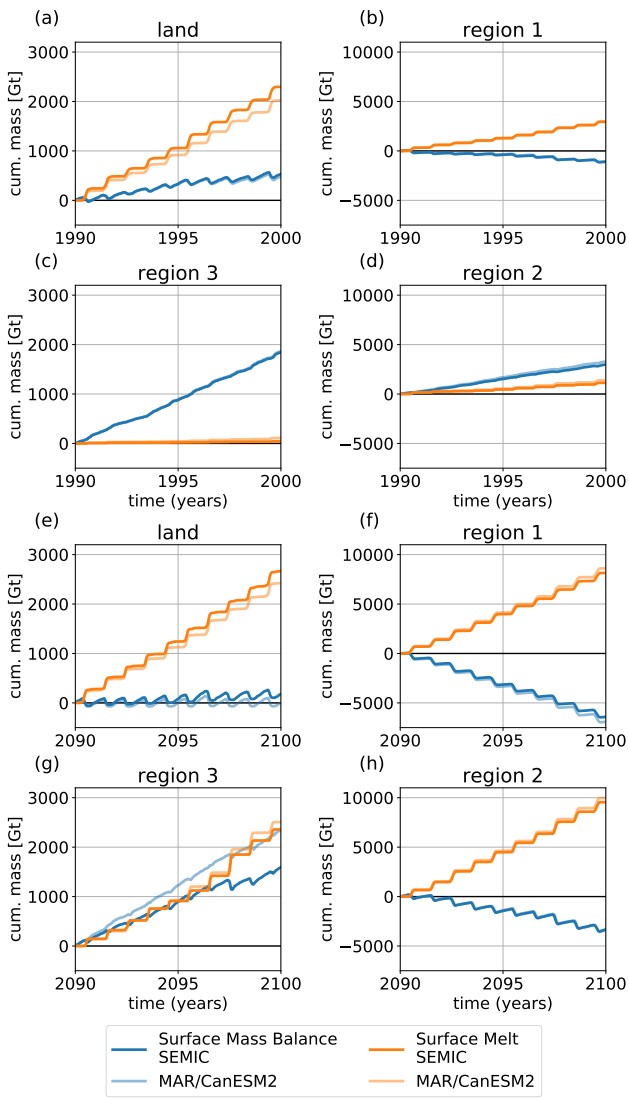

**Figure 7.** Cumulative sum of surface melt and surface mass balance over the four different regions as defined in Fig. 2 and both calibration periods, 1990–1999 (a)–(d) and 2090–2099 (e)–(h). Note the different y-scale for the land/region 3 (a), (c), (e), and (g) and for region 1/region 2 (b), (d), (f), (h).

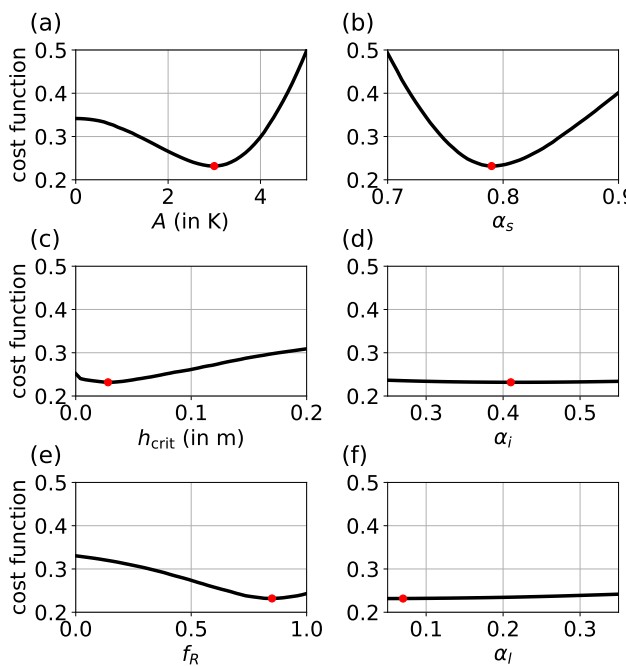

**Figure 8.** Sensitivity of cost function $J$, Eq. (20), for each of the free model parameters listed in Table 3: The diurnal cycle amplitude $A$ (a), the snow albedo $\alpha_s$ (b), the critical snow height $h_{\text{crit}}$ (c), the bare ice albedo $\alpha_i$ (d), the refreezing correction parameter $f_R$ (e), and the bare land albedo $\alpha_l$ (f). The red dot in each plot indicates the optimum as obtained by the calibration, i.e., the particle swarm optimisation.

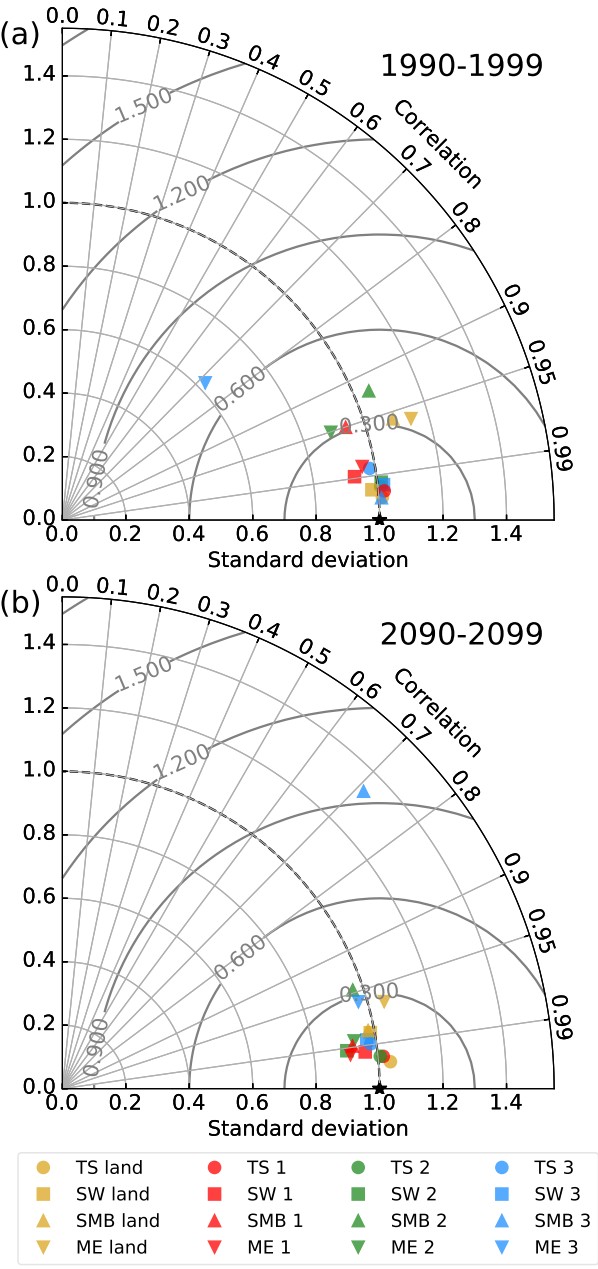

**Figure 9.** Taylor diagram of normalised surface temperature (TS), net shortwave radiation (SW), surface mass balance (SMB), and surface melt (ME) averaged over the whole Greenland ice sheet (as in Fig. 5 and 6) for the historical period 1990–1999 (a) and for the RCP8.5 scenario, years 2090–2099 (b). The black star denotes the reference field, which has (per definition) a standard deviation and a correlation coefficient of 1.

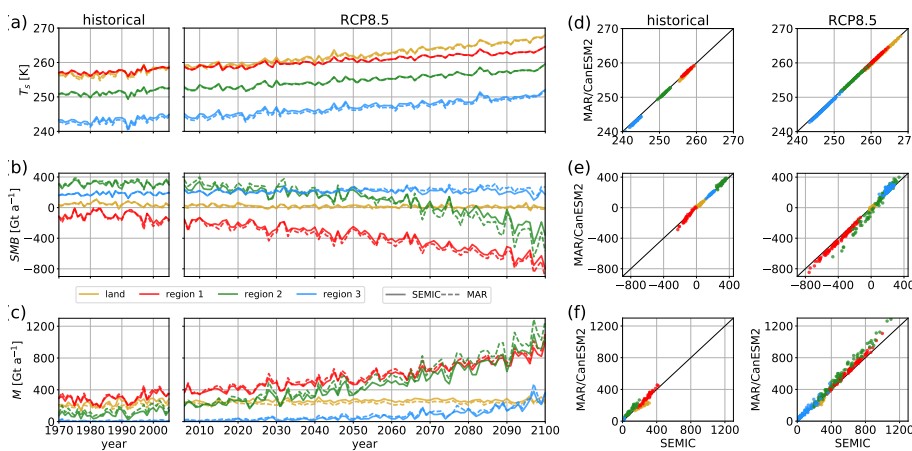

**Figure 10.** Annual-mean, region-averaged surface temperature $T_s$ (a), surface mass balance $SMB$ (b), and surface melt $M$ (c) for SEMIC (solid lines) and MAR (dashed lines) using the optimal parameter values from Table 3. Point-to-point comparison of the two models (d)–(f); variables and units as in the left panel (a)–(c).

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
