# Peer review of "SEMIC: An efficient surface energy and mass balance model applied to the Greenland ice sheet"

_The Cryosphere, 2016_

## Referee Comment (RC1) · Anonymous Referee #1 · 20 Nov 2016

This is an excellent paper on a new type of energy and mass balance (SEMIC) model that has been developed for use over the Greenland Ice Sheet. The model uses a single-layer snowpack, which is much simpler than a multi-layer scheme, but includes a realistic treatment of albedo and a parameterisation of diurnal temperature variation to account for it being run at a daily (rather than higher) time resolution. It is therefore computationally efficient compared with a fully-fledged surface energy balance and snow model, yet produces results for historical and future GrIS evolution (years 1970-2100) that are very similar to - and essentially comparable with - MAR. SEMIC is based on a model scheme that has already been used to study glacial cycles and is argued to be inherently consistent with a variety of different climatic states - although one of its key parameterisations (daily temperature cycle) will likely change under different climate states - but this can be re-tuned more readily than the positive degree-day

(PDD) factors in a PDD model, for example. Also the model code has been made open source and so is readily available to other researchers. Aspects of the model, such as the diurnal temperature variation scheme, can be significantly refined but this is for further work - the SEMIC model is already producing some excellent results. Importantly, while including realistic depiction of surface energy and mass fluxes on sub-diurnal to interannual timescales, SEMIC can be run for periods of thousands of years, for example being used as part of glacial-climate-Earth system simulations of glacial-interglacial cycles. SEMIC is methodologically superior to the PDD model which is currently widely still used for this purpose; yet SEMIC still requires downwelling shortwave and longwave radiation fields and surface wind speed, which the PDD model does not need as input. Therefore output from SEMIC is only as good as the reliability of these additional input data. Nevertheless, SEMIC presents an exciting new tool for modelling surface mass balance changes of the GrIS in a relatively computationally efficient way, so this paper should therefore be of broad interest to the glaciological, climate and modelling communities.

The manuscript is generally well argued, structured and presented and I have only a relatively few, mainly minor, specific queries and points of clarification:

page 3, line 19: rho of w (density of water) should be defined here. p.3, l.26 "For faster computation..." - faster than what? Please clarify. p.5, l.15: please add comma after "A set to zero". p.6, l.3 water density is defined here but should be defined earlier on p.3 (see above). p.6, l.18 "We neglect refreezing of melted ice and trreat ice melt as runoff." - what is the basis of this assumption? Is it reasonable and realistic for the GrIS? Adding a sentence or two of justification here would be helpful. p.7, l.8: "Tmin is set to 263.15K as originally proposed" - How reasonable is this assumption and is it supported by in situ and/or satellite data? What is the sensitivity of model results to varying it by several degC plus and minus? p.8, l.1 reword to "we refrain FROM USING..." p.8, l.15 -> "are close TO their expected trajectories." p.8, l.24: -> "while also allowing THE ASSESSMENT OF variables with different units." p.10, ll.4/5:

"While melting over the northern part of the ice sheet is overestimated by SEMIC, it is underestimated over the southern part of the ice sheet" - this seems opposite to what I interpret from studying Figure 3 - please check. p.11, l.8: -> "However, the surface mass balance itself is less sensitive TO A than melting." p.19, Figure 3 caption -> "The outlined contourS SHOW the boundaries..."
* * *

---

## Referee Comment (RC2) · X. Fettweis (Referee) · 8 Dec 2016

This paper presents a new simple energy balance snow model (called SEMIC) in the aim of simulating the GrIS SMB for forcing ice sheet models. The parameters of this model are calibrated with outputs of the regional climate model MARv2 forced by CanESM2 (RCP85). This paper is well written, fits well with TC and deserves to be published after some revisions. The physics used in SEMIC are well explained and justified. The model calibration by minimising the cost function (Figs 6 and 7) is original and statistically more robust than a simple inter-comparison. Finally, the open source mind of the authors needs to be highlighted.

However, before publication in TC, several major issues should be resolved or at least discussed:

[Figure]

Interactive
comment
* * *
1. Reference to MAR in the text

After a quick read of this paper, it appears that SEMIC is comparable to MAR !! (e.g. abstract line 7; conclusion: line 9). But claiming this, is lie to the readers because SEMIC is comparable in fact to the snow model used in MAR (daily MAR atmospheric outputs were used to force SEMIC). The MAR snow model, based on the CROCUS snow model and called SISVAT, uses exactly the same input (precip, temperature, radiative flux, humidity, wind, surface pressure) than SEMIC. The only difference between SEMIC and SISVAT is the forcing time step (150s in SISVAT vs daily in SEMIC). Therefore, it is normal for example that the interannual/daily variability is very well correlated with the MAR outputs as the variability of the snowpack is mainly driven by the near-surface atmosphere variability and precipitation.

- First, all the inter-comparisons of Surf Temp. and SMB components with MAR must refer to SISVAT (used in the MAR model) and not to MAR!

E.g. : pg 10 line 19: "The Taylor diagram in Fig. 7 summarises the performance of SEMIC compared to MAR and its multi-layer snowpack model" MUST BE "The Taylor diagram in Fig. 7 summarises the performance of SEMIC compared to the multi-layer snowpack model used in MAR". The meaning of the corrected sentence is fully different.

- Secondly, SEMIC does not allow to take into account the atmosphere-snowpack interactions. But, as it is forced by MAR, theses feedbacks (and notably the albedo feedback) are taken into account here. This should be mentioned in the manuscript.

- Thirdly, it is true that MAR is very slow in respect to SEMIC but it is not due to its snow model which is fully parallelized (taking about 5 % of its computing time) but it is due to the physical atmospheric downscaling. As next step, it will be a lot of more interesting to compare SEMIC forced directly by CanESM2 vs SISVAT-MAR forced by CanESM2 but it is clearly out of scope of this paper. Such comparison has been made

by Geyer et al. (TCD, 2013) using CROCUS as snow model. They showed well that the biggest issue is not the snow model but the downscaling of the atmospheric fields (e.g. precipitation). This should be mentioned in the manuscript. Finally, the SISVAT snow model as well as the raw CROCUS snow model can be run in stand alone mode like SEMIC. Therefore, this shows well that this paper is well SEMIC vs CROCUS and not SEMIC vs MAR.

Geyer, M., Salas Y Melia, D., Brun, E., and Dumont, M.: The Greenland ice sheet: modelling the surface mass balance from GCM output with a new statistical downscaling technique, The Cryosphere Discuss., 7, 3163-3207, doi:10.5194/tcd-7-3163-2013, 2013.

—————————————————————————————————————————————————————

2. Calibration with MAR outputs only

- Firstly, while this model is very fast, I am a bit surprised that the calibration was only made over three years (2098-2100). I understand the aim of being able to simulate extreme values. But 3 yrs is very short and a validation (as Figure 7) over both 10 yrs periods (2090-2100 and 1990-2000) will be more robust. As nothing is linear, using current climate (which is the only climate we know well) can be also useful. If it is not a big job for the authors, I will like to see the sensitivity of the extreme values of the parameters from Table 1 over these both periods.

- Secondly, I am also surprised that the sensitivity of the bare ice albedo value is not tested. In MAR, this one is the more sensible parameters (as explained in Fettweis et al., 2016). As SEMIC underestimates melt in respect to MAR, lower values of bare ice albedo can reduce this bias.

Fettweis, X., Box, J. E., Agosta, C., Amory, C., Kittel, C., and Gallée, H.: Reconstructions of the 1900–2015 Greenland ice sheet surface mass balance using the regional climate MAR model, The Cryosphere Discuss., doi:10.5194/tc-2016-268, in review,

2016.

- Thirdly, the parameters of SEMIC are calibrated with MAR outputs only. But MAR is not the true!! and as shown in Fettweis et al. (2016), MARv2 (used here) overestimates the melt in respect to MARv3.5.2. A calibration/validation over current climate using the SMB PROMICE data set will be more robust. Outputs of MAR forced by NCEP1 or ERA can be used as forcing for this. It is clear that current climate is not future climate but the melt rate of the recent summers is already significant. I don't ask to recalibrate SEMIC over current climate using the SMB PROMICE data set but this issue should be at least mentioned in the manuscript. I can provide daily outputs of MARv3.5.2 forced by CanESM2 (rcp85) to check the sensitivity of the calibration to the used MAR version.
* * *
3. Cumulated SMB change.

SEMIC was built to force an ice sheet model but an ice sheet model is not sensitive to the daily variability and in a less extent to the interannual variability of SMB. It is mainly sensitive to the cumulated SMB changes. When we are looking on Fig 8, SEMIC seems to diverge from MAR after 2050. What are the total cumulated differences in 2100? For me, the calibration should be made to have the same cumulated SMB changes over 2000-2100 than MAR and not to have good results over 2098-2100 only. Due to error compensations, having a too high/too low SMB several years in respect to MAR is not a problem for an ice sheet model which will give the same results at the end than if it will be forced by MAR. The best will be to calibrate SEMIC over current climate when we have other estimations of cumulated SMB changes than MAR (van den broeke et al., TC, 2016).
* * *
Minor remarks:
- Fig 2: the SMB zones shown in Fig 2 were only valid over the 1990's (based on Zwally and Giovineto, JGR, 2002) and were formerly used in MARv2 to initialise the snow model. These boundaries are already no more relevant for current climate of the 2000's. This issue should be mentioned in the manuscript.

Fig3: MAR/ERA-40 must be SISVAT/CanESM2. It should be interesting to show the differences over current climate (in supplementary material) when MAR is forced by reanalysis. This error in the legends means that such a comparison has already been done.

Fig. 5: not useful => supplementary material.

Fig. 8: showing an equivalent of Fig 4 with cumulated values will be more useful.

---

## Author Response (AR2)

**Authors' Response to Anonymous Referee #1**

We are grateful to the referee for reading our manuscript and we thank for her/his comments.

In the following we reply to each of the referee's comments. We highlight individual parts of the comments that we are going to address in italics. Our response is put below each comment together with our proposed changes to the manuscript; where these changes will appear in the revised manuscript is put in parentheses.

- *page 3, line 19: rho of w (density of water) should be defined here.*
  - Yes, we need to define $\rho_w$ here and will do so in the revision.(page 3, line 19)
- *p.3, l.26 "For faster computation..." - faster than what? Please clarify.*
  - We meant faster computation because of the daily time step. Of course this is relative, so we will delete that phrase and instead simply state: *We use a time step of one day.* (p.3, l.26)
- *p.5, l.15: please add comma after "A set to zero".*
  - We will add a comma to clarify that sentence. (p.5, l.15)
- *p.6, l.3 water density is defined here but should be defined earlier on p.3 (see above).*
  - In the revised manuscript, $\rho_w$ should be already defined before as mentioned in the first bullet point.
- *p.6, l.18 "We neglect refreezing of melted ice and treat ice melt as runoff." - what is the basis of this assumption? Is it reasonable and realistic for the GrIS? Adding a sentence or two of justification here would be helpful.*
  - Ice itself does not retain any meltwater at the surface and we assume that it has a water holding capacity of effectively zero. If a snow pack is present we assume that it can retain melt water. However, it turns out that we have missed adding a refreezing fraction $f_R$ to the rhs of Eq. (16c), which reduces the potential refreezing according to this parameter. In the revised manuscript $f_R$ is one of the free model parameter and included in the parameter calibration (varying between 0 and 1).
  - In the revised manuscript Eq. (16c) should read: $R = R_{\text{rain}} + R_{\text{melt}} = f_R(R_{\text{pot,rain}} + R_{\text{pot,melt}})$ with $f_R$ being the refreezing fraction and a free model parameter.
  - To explain the neglect of refreezing of melted ice, we suggest the two sentence from above: Ice itself does not retain any meltwater at the surface and we assume that it has a water holding capacity of effectively zero. If a snow pack is present we assume that it can retain melt water. (to be added to p.6, l.18)
- *p.7, l.8: "Tmin is set to 263.15K as originally proposed" - How reasonable is this assumption and is it supported by in situ and/or satellite data? What is the sensitivity of model results to varying it by several degC plus and minus?*
  - After revising the albedo parameterisation we decide not to use the proposed approach by Slater et al. (1998). We find that $\alpha_{s,\max}$ and $\alpha_{s,\min}$ are only a couple of per cent apart of each other, (0.77 and 0.80, respectively in the submitted manuscript, see Table 1). This means that the overall effect of this parameterisation will be at max 3 per cent, which is not much added value in our opinion. Instead, we decided to reduce the complexity of the albedo parameterisation and the number of free model parameters. Hence, $\alpha_{s,\min}$ and $T_{\min}$ are no longer needed. However, for other (future) application we keep the albedo in the model to be chosen by the user optionally but will not mention it in the manuscript to avoid confusion.
  - We propose to simplify Sect. 2.4 (Snow albedo parametrisation) in the manuscript. The albedo parameterisation simplifies to $\alpha = \alpha_s - f_a(\alpha_s - \alpha_{bg})$, i.e., Eq. (19). Eq. (18a,b) are no longer needed. Changes in Sect. 2.4 and throughout the revised manuscript will be made accordingly. (Sect. 2.4, p7-8)
- *p.8, l.1 reword to "we refrain FROM USING..."*
  - We will revise this part of the manuscript because of justified comments by the second referee, who suggests to use more than three years for the calibration to make a more robust parameter estimation. (p.8 ll.1-4)
- *p.8, l.15 -> "are close TO their expected trajectories."*
  - We will change the sentence as proposed.
- *p.8, l.24: -> "while also allowing THE ASSESSMENT OF variables with different units."*

- We will change the sentence as proposed.
- *p.10, ll.4/5: "While melting over the northern part of the ice sheet is overestimated by SEMIC, it is underestimated over the southern part of the ice sheet" - this seems opposite to what I interpret from studying Figure 3 - please check.*
    - No, the difference between SEMIC and MAR is positive in the northern part and negative in the southern part of the ice sheet. Here, melt is defined as positive quantity, although the loss of mass by melt is negative in a physical sentence. But apparently this is be confusing so we suggest to add a sentence here. For example this might be helpful:
    - Note that melt is defined as a positive quantity but is subtracted from the surface mas balance (p.10 l.5)
- *p.11, l.8: -> "However, the surface mass balance itself is less sensitive TO A than melting."*
    - We will change the sentence as proposed.
- *p.19, Figure 3 caption -> "The outlined contourS SHOW the boundaries..."*
    - We will change the sentence as proposed.

Mario Krapp (on behalf of the authors)

**Authors' Response to Xavier Fettweis**

We thank Xavier Fettweis for his helpful comments and for the pointing out the clear distinction between MAR and its snowpack model. Xavier raised some valid points which we would like to address in our revised manuscript.

In the following we reply to each of the referee's comments. We highlighted individual parts of the comments that we are going to address in italics. Changes in the manuscript have been described and highlighted in italics; where these changes appear in the updated manuscript is put in parentheses.

**1. Reference to MAR in the text**

- *[. . . ] SEMIC is comparable in fact to the snow model used in MAR [. . . ]* **and** *[. . . ] all the inter-comparisons of Surf Temp. and SMB components with MAR must refer to SISVAT (used in the MAR model) and not to MAR!*
    - We acknowledge that MAR is more than just the surface energy and mass balance model but we think that we already differentiate between SEMIC and MAR right in the beginning. For example, in the abstract we write that SEMIC is able *to reproduce surface characteristics and day-to-day variations similar to the regional climate model MAR* and that SEMIC is *in good agreement with the more sophisticated multi-layer snowpack model included in MAR*. For the sake of clarity and understanding we simply use the model name MAR whenever we refer to the surface snowpack model (i.e., SISVAT). However, to give reasonable credit to the MAR community should add a brief description what we exactly compare our model to.
    - We propose to add a brief description of SISVAT and its part in the MAR framework in Sect 2.5 (page 7). However, throughout the paper we are still simply refering to MAR output whenever a comparison to SEMIC is made.
- *Secondly, SEMIC does not allow to take into account the atmosphere-snowpack interactions. But, as it is forced by MAR, theses feedbacks (and notably the albedo feedback) are taken into account here. This should be mentioned in the manuscript.*
    - That is true in the sense that atmospheric characteristics are prescribed and are not affected by surface properties simulated by SEMIC. This is because in this work we used MAR output for testing of SEMIC. However, since SEMIC calculates surface-atmosphere heat fluxes such latent heat, sensible heat, and upward longwave radiation, in the future it is planned to coupled SEMIC bi-directionally with an appropriate atmosphere model. Then atmosphere-snowpack interaction will be properly accounted for.
    - However, we will add a brief explaination that the current setup only serves the purpuse of mimicking the response of SEMIC to an atmosphere forcings but that the surface response cannot be not taken into account by the atmosphere.
- *Thirdly, it is true that MAR is very slow in respect to SEMIC [. . . ] due to the physical atmospheric downscaling. [. . . ] This should be mentioned in the manuscript.*
    - We do not want to blame MAR in terms of runtime. But explaing why a regional climate model needs more computational time is not within the scope of our paper. However, we can make clear that SEMIC and MAR are two different classes of models and we will add sentence about the distinction in the part above (Sect. 2.5).
- *Finally, the SISVAT snow model as well as the raw CROCUS snow model can be run in stand alone mode like SEMIC. Therefore, this shows well that this paper is well SEMIC vs CROCUS and not SEMIC vs MAR.*
    - We use the term MAR for simplicity but will stick to it, also because SISVAT is a component of the MAR model (see our first bullet point).

**2. Calibration with MAR outputs only**

- *I am a bit surprised that the calibration was only made over three years [. . . ]. But 3 yrs is very short and a validation [. . . ] over both 10 yrs periods (2090-2100 and 1990-2000) will be more robust.*
    - We agree that a longer calibration period over both, present-day and future-warming conditions is more robust and would lead to more robust optimal paramter estimate. We changed the forcing data for the proposed periods, 1990-2000 and 2090-2100, but because of the computational overhead we used just a subset of the MAR data, i.e., a random sub-sample accounting for 20% of land and ice points.
    - The calibration procedure will be thouroughly revised and extended in Sect. 3, Model parameter calibration (page 8).
- *[. . . ] the sensitivity of the bare ice albedo value is not tested. In MAR, this one is the more sensible parameters (as explained in Fettweis et al., 2016). As SEMIC underestimates melt in respect to MAR, lower values of bare ice albedo can reduce this bias.*
    - Yes, the bare ice albedo has not been tested. We will add the sensitivity of bare ice albedo $\alpha_i$, as well as of bare land albedo $\alpha_l$, and the missed $f_R$ to Sect. 3.4, Parameter Sensitivity (p. 10-11). Figure 6 will be also updated accordingly (page 22).
    - Note, that we will no longer use the Slater et al. (1998) albedo parameterisation (for reasons, see our response to referee #1). Therefore, $\alpha_{s,\min}$ will no longer be part of the sensitivity in Figure 6 (page 22) and the text (p. 10-11).
- *MARv2 [. . . ] overestimates the melt in respect to MARv3.5.2. A calibration/validation over current climate using the SMB PROMICE data set will be more robust. [. . . ] I don't ask to recalibrate SEMIC over current climate using the SMB PROMICE data set but this issue should be at least mentioned in the manuscript.*
    - We know that with each model update several aspects of that model are being improved. However, with this paper we show that SEMIC can be a surrogate of a sophisticated snow-pack model and we prove it. The parameters will likely change if forced with different data and calibrated for different data.
    - We will add a brief discussion of this issue in Sect. 5, Discussion (p. 13-14).

**3. Cumulated SMB change**

- *SEMIC seems to diverge from MAR after 2050. What are the total cumulated differences in 2100? For me, the calibration should be made to have the same cumulated SMB changes over 2000-2100 than MAR and not to have good results over 2098-2100 only. Due to error compensations, having a too high/too low SMB several years in respect to MAR is not a problem for an ice sheet model which will give the same results at the end than if it will be forced by MAR. The best will be to calibrate SEMIC over current climate when we have other estimations of cumulated SMB changes than MAR (van den broeke et al., TC, 2016).*
    - Yes, SEMIC diverges from MAR in the RCP8.5 scenario. We aim to be as close to MAR results as possible. But we cannot use the whole period of MAR data (i.e, 1970-2100) to calibrate our model. First, we need to strictly differentiate between training data and test data. We do so be defining the periods 1990-2000 and 2090-2100 as our training data. Second, it is computationally not feasible to use all MAR model years for the parameter calibration, which needs to be run several tens of thousands of times.
    - There is a valid point in this comment. That is to quantify the differences between SEMIC and MAR over time, i.e., the cumulative difference. For that reason we can add a figure, similar to Fig. 4, but showing the cumulative differences of mass balance terms over time, e.g., for the whole period 1970-2100 or for the historical and RCP8.5 period separately.

**Minor Remarks**

- *Fig 2: the SMB zones shown in Fig 2 were only valid over the 1990's [. . . ]. These boundaries are already no more relevant for current climate of the 2000's. This issue should be mentioned in the manuscript.*
  - We use the zones only to differentiate the major climatic regimes across the Greenland Ice Sheet. For example, on page 9, ll.3 we point out that the *regions crudely represent the main ablation zones at the ice-sheet margins (region 1), the main accumulation zone at ice-sheet interior.* However, we will remark here that the regions only represent different SMB zones for today's climate and may not be valid for any future warming scenario such as RCP8.5. (Sect 3.1, p. 9)
  - The only distinction *SEMIC* makes is between ice-covered and ice-free zones, i.e., land/ice mask. We will also add a remark to Fig. 2 to explain that the differentation is only valid for present day. (page 18)
- *Fig 3: MAR/ERA-40 must be SISVAT/CanESM2. It should be interesting to show the differences over current climate (in supplementary material) when MAR is forced by reanalysis. This error in the legends means that such a comparison has already been done.*
  - Indeed, *MAR/ERA-40* output has been used at an earlier stage of our research. However, to minimize the number of different datasets used for that study, we decided to solely focus on *MAR/CanESM2* output for our calibration and validation process. The error in the legend has been changed accordingly.
- *Fig. 5: not useful => supplementary material.*
  - We disagree, while the maps in Fig. 3 provides a visual for the spatial (but ime-averaged) differences between MAR and SEMIC, Fig. 5 provides a visual for the temporal differences between the two models. Having both figures in the main text will provide a more cokplete picture of SEMIC and MAR, both in space and time.
- *Fig. 8: showing an equivalent of Fig 4 with cumulated values will be more useful.*
  - This will be shown in a new figure that depicts the cumulative differences between SEMIC and MAR (see our last bullet point in the comment section 3, cumulated SMB changes).

Mario Krapp (on behalf of the authors)

**Response to the Editor**

- *1) I would like to ask you to consider, if possible, to add something to the title that suggests that the model here proposed is calibrated using MAR outputs. This should also be stressed (with this aspect more important than the title) in the text, especially in the discussion and conslusions sections. The outptus of the regional climate model used to calibrate the model are heavily dependent on the version used (here 3.2) and the forcing. The authors need to highlight this and, particularly, that the model can reproduce MAR outputs but care should be taken to claim the validitity of SEMIC as presented in this study. Also, it would be important for the authors to suggest how the model can be used in a more general way and what would be the impact of using different MAR (or RACMO ?) versions on the results.*
    - We agree that the contribution of Xavier Fettweis and the MAR team should be acknowledged thoroughly. We explain in the beginning of the discussion how valuable the publicly available MARv2 data are and, as you suggested, that MARv2 is somehow outdated and any results from SEMIC should be taken with a grain of salt
    - We also add to the conclusion that SEMIC has been forced with atmospheric fields from MARv2 and our comparison is mainly done with its surface model SISVAT (also suggested by Xavier in his review)
- *The other point is that the authors offer no validaiton of the results using in-situ measurements. Is this something that can be at least discussed ?*
    - Actually, we did a preliminary analysis (before submitting the paper) for which SEMIC has been forced by meteorological data (Morin et al., 2012) but we didn't want to include this analysis in the first place because i) we think it would make the paper even more complicated and bloated, and ii) that the reader might miss the point of describing SEMIC in detail. We think a comprehensive comparison using SEMIC with different types of climate datasets, e.g., other regional climate models (or MAR versions), re-analysis data, or other in-situ observations (e.g., PROMICE or GC-NET) would be a nice follow-up study.
    - We now mention in the discussion that we did a preliminary analysis (with the Morin et al. (2012) dataset) but provide no further details.
- *2) the second important point is that the paper lacks of appropriate references. I would strongly encourage the authors to update them and increase their number. This should be done throughout the paper. As an example (though , again, the authors should check through the whole paper), in Section 3 there is no justification or reference for choosing the snow height to 1 m. Also, references to previous work concerning snow energy balance should be added together with those referencing to the origin of equations used and the choices made (as , for example, the case mentioned above). again, this should be properly done throughout the paper.*
    - We agree and add appropriate references where needed, namely in the discussion, the model setup section, and in the discussion
    - Additional references:
        * Franco et al.: Future projections of the Greenland ice sheet energy balance driving the surface melt, The Cryosphere, 7, 1–18, 2013.
        * Fettweis et al.: Reconstructions of the 1900–2015 Greenland ice sheet surface mass balance using the regional climate MAR model, The Cryosphere Discussions, 2016, 1–32, 2016.
        * van As et al.: Placing Greenland ice sheet ablation measurements in a multi-decadal context, Geological Survey of Denmark and Greenland Bulletin, 35, 71–74, 2016.
        * Noel et al.: Evaluation of the updated regional climate model RACMO2.3: summer snowfall impact on the Greenland Ice Sheet, The Cryosphere, 9, 1831–1844, 2015.
        * Dee et al.; The ERA-Interim reanalysis: configuration and performance of the data assimilation system, Quarterly Journal of the Royal Meteorological Society, 137, 553–597, 2011.
        * Morin et al.: An 18-yr long (1993–2011) snow and meteorological dataset from a mid-altitude mountain site (Col de Porte, France, 1325 m alt.) for driving and evaluating snowpack models, Earth System Science Data, 4, 13–21, 2012.
        * Oerlemans, J.: The mass balance of the Greenland ice sheet: sensitivity to climate change as

revealed by energy-balance modelling, The Holocene, 1, 40–48, 1991.

* Oerlemans, J. and Knap, W.: A 1 year record of global radiation and albedo in the ablation zone of Morteratschgletscher, Switzerland, Journal of Glaciology, 44, 231–238, 1998
* Bougamont et al.: Impact of model physics on estimating the surface mass balance of the Greenland ice sheet, Geophysical Research Letters, 34, L17 501, 2007.
* Greuell et al.: Modelling land-ice surface mass balance, p. 117–168, Cambridge University Press, 2004.

- *3) Figures dont have labeling for each panel. These should be added and captions should be changed accordingly. figures 5 and 6 are not clearly legible. It is impossible to separate between the MAR- and SEMIC-modeled quantities. Authors should improve these figures. Figure 7 is missing units on the left panels. Also, units are usually reported in square parentheses. Figure 10: 'lighter colours' is not a good reference for readers. Authors should include a label or similar to indicate the different quantities associated with the colors.*
  - We add labels to Figures 1, 3, 4, 7, 8, 9, and 10.
  - We hope that Figures 5 and 6 are better legible as we increased font size and use slightly transparent and thinner lines.
  - Units now appear in squared brackets.
  - Figure 10 has distinguishes SEMIC and MAR using different line styles, with appropriate labels added.

Mario Krapp (on behalf of the authors)

**List of all relevant changes**

Changes as requested by the reviewers:

- We mention the snowpack model SISVAT in the abstract and in the Model Setup (Sect. 2.5)
- We add a sentence to specify that air density is not computed by MAR but derived from the ideal gas law
- We add the model parameter $f_R$ into the model equations (Sect. 2.3) and throughout the manuscript
- The snow albedo parameterisation has been simplified (Sect 2.4)
- The model setup was rewritten to account for the different (historical and RCP8.5) and longer calibration period (Sect 2.5) and a slightly different calibration setup
- Accordingly, the new calibration periods have been added to the Model calibration (Sect 3)
- An error in the centred root mean square definition has been corrected (Eq. 19)
- The model calibration setup led to a new optimal parameter set, which has been updated in Sect 3.1 and Table 3
- Characteristic ice-sheet variables and differences to MAR have changed because of the new optimal parameter set and numbers have been updated in Sect 3.3 and Table 4
- The parameter sensitivity has been simplified and only accounts for the overall cost function $J$; Sect 3.4 has been rewritten therefore
- The discussion has been updated to reflect the major changes in model setup and its calibration
- Maps and plots which showed the single previous calibration period do now show both periods, 1990-1999 and 2090-2999: Figs 3 & 4, Figs. 5 & 6, Fig. 7, Fig. 9

Changes as requested by the editor:

- We explain in the discussion that MARv2 is outdated and any results from SEMIC should be taken with a grain of salt
- We add to the conclusion that SEMIC has been forced with MARv2 atmospheric fields and that the comparison has been done with MAR's surface module SISVAT
- We add a sentence about SEMIC and its potential application to different types of climatic data, e.g., re-analysis or in-situ observations and that we did some preliminary analysis with meteorological forcing data from Col de Porte (Morin et al., 2012)
- We add 10 additional reference in the introduction, the model setup, and the discussion to increase the actual number of references and to support some claims we made throughout the respective parts of the manuscript
- We add appropriate labels to the figures
- Units are displayed in square brackets
- We improved figure 5 and 6 to increase their readability
- We changed the line style for MAR in Fig. 10 and add an appropriate label to the figure

[revised manuscript text omitted]

is as follows: Albedo declines if snow starts to melt and melting is much more likely for higher temperatures. The snow albedo above a certain temperature threshold, here $T_{min}$, is temperature dependent and starts to decline to theof old snow, i.e., $\alpha_{s,\min}$

$$\alpha_s \quad = \alpha_{s,\max} - (\alpha_{s,\max} - \alpha_{s,\min}) t_m^3 \quad \text{with}$$

$$t_m \quad = \begin{cases} 0 & \text{if } T_s < T_0, \\ \frac{T_s - T_{\min}}{T_0 - T_{\min}} & \text{if } T_{\min} \leq T_s < 
[revised manuscript text omitted]

$c_{\text{eff}}$

$C_S$

$C_L$

$c_{p,a}$

$\sigma$

$T_0$

$\rho_w$

$L_s$

$L_v$

$L_m$

$T_{\min}$ 263.15 K minimum temperature threshold for albedo parametrisation$h_{s,\max}$

$\alpha_I$ 0.45 bare ice albedo, i.e., clean or blue ice $\alpha_l$ 0.15 bare land albedo height

**Table 1.** Model constants and their description.

| symbol | description |
|---|---|
| $SW^{\downarrow}$ | downwelling shortwave radiation  [W m$^{-2}$] |
| $LW^{\downarrow}$ | downwelling longwave radiation  [W m$^{-2}$] |
| $\rho_a$ | air density  [kg m$^{-3}$] |
| $u_s$ | surface wind speed  [m s$^{-1}$] |
| $T_a$ | near-surface air temperature  [K] |
| $q_a$ | near-surface specific humidity  [kg kg$^{-1}$] |
| $p_s$ | surface pressure  [Pa] |
| $P_s$ | snowfall rate  [m s$^{-1}$] |
| $P_r$ | rainfall rate  [m s$^{-1}$] |

**Table 2.** Atmospheric forcing fields needed as input for this model.

| symbol | range | value | description |
|---|---|---|---|
| $A$ | 0.0–5.0 | **3.0** | amplitude of diurnal cycle [K] |
| $\alpha_s$ | 0.70–0.90 | **0.79** | fresh dry snow albedo |
| $\alpha_i$ | 0.25–0.55 | **0.41** | bare ice albedo, i.e., clean or blue ice |
| $\alpha_l$ | 0.05–0.35 | **0.07** | bare land albedo |
| $h_{\mathrm{crit}}$ | 0.00–0.20 | **0.028** | critical snow height for albedo parameterisation [m] |
| $f_R$ | 0.0–1.0 | **0.85** | refreezing correction |

**Table 3.** Model parameters with their initial range and their optimal value in bold face.

**Table 4.** Comparison of SEMIC and MAR. Shown are multi-year mean averages over the ice sheet (regions 1–3) and ice-free land, their mean gridpoint-to-gridpoint differences $\overline{\Delta}$, their minimum, and their maximum gridpoint-to-gridpoint differences, $\min\Delta$ and $\max\Delta$. Here, ice sheet means all ice-covered regions (region 1–3).

| | | 1990–1999 | | | | | 2090–2099 | | |
| --- | --- | --- | --- | --- | --- | --- | --- | --- | --- |
| | | SEMIC | MAR | $\overline{\Delta}$ | $\min\Delta$ | $\max\Delta$ | SEMIC | MAR | $\overline{\Delta}$ |
| ice sheet | $T_s$ [K] | 249.6 | 249.2 | 1.4 | 0.2 | 4.8 | 256.1 | 255.8 | 1.3 |
| | $SMB$ [mm day$^{-1}$] | 1.57 | 1.61 | 0.96 | -1.78 | 2.88 | -0.24 | -0.21 | 0.97 |
| | $M$ [mm day$^{-1}$] | 1.62 | 1.68 | 0.94 | -0.79 | 3.68 | 4.05 | 4.20 | 0.84 |
| | $SW_{net}$ [W m$^{-2}$] | 28.7 | 27.7 | 1.9 | -10.9 | 14.2 | 31.9 | 32.0 | 0.9 |
| land | $T_s$ [K] | 258.4 | 257.9 | 1.5 | -0.1 | 5.1 | 267.5 | 267.3 | 1.2 |
| | $SMB$ [mm day$^{-1}$] | 1.27 | 1.25 | 1.03 | 0.67 | 1.56 | 1.09 | 1.00 | 1.09 |
| | $M$ [mm day$^{-1}$] | 2.18 | 2.04 | 1.14 | 0.60 | 1.79 | 2.37 | 2.25 | 1.12 |
| | $SW_{net}$ [W m$^{-2}$] | 46.8 | 47.3 | 0.4 | -20.7 | 22.6 | 61.7 | 65.6 | -2.9 |

Comparison of SEMIC and MAR. Shown are multi-year (2098–2100) mean averages over the ice sheet and ice-free land, their mean difference, and the minimum and maximum differences. Compare also to Fig. **??**.

[revised manuscript text omitted]